# Pretraining with Re-parametrized Self-Attention: Unlocking Generalization in SNN-Based Neural Decoding Across Time, Brains, and Tasks

**Yuqi Yang**[1]*, **Tengjun Liu**[1,2]*, **Haiyan Zhang**[1], **Ruixue Wang**[3], **Xuchao Chen**[1],
**Mingkang Li**[1], **Yansong Chua**[4]†, **Nenggan Zheng**[1], **Shaomin Zhang**[1]†
[1] Zhejiang University, Hangzhou, China
[2] Intellizon Medical Ltd., Suzhou, China
[3] Westlake University, Hangzhou, China
[4] Scalefree Intelligence Ltd., Shanghai, China

## Abstract

The emergence of large-scale neural activity datasets provides new opportunities to enhance the generalization of neural decoding models. However, it remains a practical challenge to design neural decoders for fully implantable brain-machine interfaces (iBMIs) that achieve high accuracy, strong generalization, and low computational cost, which are essential for reliable, long-term deployment under strict power and hardware constraints. To address this, we propose the Re-parametrized self-Attention Spiking Neural Network (RAT SNN) with a cross-condition pretraining framework to integrate neural variability and adapt to stringent computational constraints. Specifically, our approach introduces multi-timescale dynamic spiking neurons to capture the complex temporal variability of neural activity. We refine spike-driven attention within a lightweight, re-parametrized architecture that enables accumulate-only operations between spiking neurons without sacrificing decoding accuracy. Furthermore, we develop a stepwise training pipeline to systematically integrate neural variability across conditions, including neural temporal drift, subjects and tasks. Building on these advances, we construct a pretrained model capable of rapid generalization to unseen conditions with high performance. We demonstrate that RAT SNN consistently outperforms leading SNN baselines and matches the accuracy of state-of-the-art artificial neural network (ANN) models with much lower computational cost under both seen and unseen conditions across various datasets. Collectively, pretrained-RAT SNN represents a high-performance, highly generalizable, and energy-efficient prototype of an SNN foundation model for fully iBMI. Code is available at RAT SNN GitHub.

## 1 Introduction

Brain-machine interfaces (BMIs) have advanced significantly in enabling direct brain-to-device communication (Liu et al., 2020; Zhang et al., 2023; Rapeaux & Constandinou, 2021). Implantable BMIs (iBMIs), recording high-fidelity neural signals via cortical microelectrode arrays, offer superior signal quality and control precision compared to non-invasive ones (Rapeaux & Constandinou, 2021). Prior work has demonstrated the feasibility of using iBMIs for autonomous robotic arm control (Hochberg et al., 2012; Flesher et al., 2021) and text generation (Pandarinath et al., 2017; Willett et al., 2021; Moses et al., 2021; Card et al., 2024). However, enabling decoding models to generalize across subjects, tasks, and time remains a core challenge. This difficulty stems from intrinsic neural variability, including inter-subject differences, inter-task variations, and temporal drift within subjects (Driscoll et al., 2022). Such variability makes it difficult to apply a single model across diverse datasets: the same task might elicit different neural patterns in different subjects, while a single subject may exhibit distinct activity under varying contexts (Dinstein et al., 2015).

---

*Equal contribution. †Corresponding authors, shaomin@zju.edu.cn, James_4424@hotmail.com.

Originally developed in natural language processing and computer vision (Wang et al., 2023), foundation models aim to scale learning across domains, improving generalization and minimize retraining. In iBMIs, such models hold promise for improving cross-condition generalizability, enabling rapid deployment when subject-specific data are scarce or costly to obtain. Recent advances in large-scale neural recording have made it possible to obtain rich datasets (Perich et al., 2018; Pei et al., 2021; O'Doherty et al., 2017; Churchland et al., 2012), making it feasible to train foundation neural decoding models that generalize across varied conditions including sessions, subjects, and tasks.

Several Artificial Neural Network (ANN) based foundation models have recently been proposed, including POYO (Azabou et al., 2023) and NDT3 (Ye et al., 2025). They share a common underlying paradigm: scaling up to accommodate large volumes of data. Yet, their reliance on large models and significant computational resources hinders real-time deployment in resource-constrained fully iBMIs, which reduce infection risk by eliminating external pedestals and also promote long-term adaptation and privacy (Yik et al., 2025; Taeckens & Shah, 2024; Yuste et al., 2017). But they impose strict constraints on decoder computational energy consumption, model size, and latency. Therefore, to meet the requirements of fully iBMIs, it is imperative to develop decoders that can deliver reliable and accurate performance under strict latency and power constraints.

Spiking neural networks (SNNs), mimicking biological spike-based event-driven (spike-driven) nature (Yao et al., 2023b), significantly reduce computational costs by replacing multiply-accumulate (MAC) with accumulate-only (AC) operations (Yin et al., 2021). Evolving in discrete state spaces, SNNs exhibit greater robustness to temporal noise (Lichtsteiner et al., 2008), and are considered advantageous for processing discrete spatiotemporal outputs (Liang et al., 2023; Wu et al., 2024). Especially, SNNs, whose event-driven and discrete nature aligns well with the spiking paradigm, are particularly compatible with cortical spike train (CST), thereby minimizing the need for extensive data preprocessing. However, existing research on SNN-based CST decoding remains relatively preliminary, and the architectures employed are often too simplistic to either achieve high-performance decoding or serve as cross-condition foundation models.

To address the above challenges of building accurate, generalizable and efficient CST decoder, we propose Re-parametrized self-Attention (RAT) SNN. It features a spike-driven self-attention enhanced with re-parameterization, allowing the use of batch normalization (BN) during training while maintaining AC-only operations between synapses during inference. In addition, RAT SNN introduces recurrent connections among leaky integrate-and-fire (LIF) neurons combined with dynamic synapses to capture multi-timescale dynamics effectively. Furthermore, our method introduces subject-specific BN as a lightweight approach to accommodate distributional shifts across subjects. Through a custom cross-condition training pipeline, RAT SNN-CC achieves robust integration across time, subjects, and tasks, yielding a scalable, low-power SNN foundation model for generalizable neural decoding. Our main contributions can be summarized as follows:

- We propose RAT SNN, a lightweight spiking neural network that integrates re-parameterized spike-driven self-attention with multi-timescale dynamics, and maintains AC-only inter-synaptic operations and architectural efficiency.
- We introduce subject-specific BN along with a stepwise pretraining framework for RAT SNN, enabling the integration of neural temporal drift within an individual subject as well as other neural variability across subjects and tasks with spike-driven inference.
- We evaluate our method on multiple datasets, encompassing diverse conditions. RAT SNN demonstrates rapid generalization, consistently outperforms state-of-the-art (SOTA) SNN baselines, and matches the performance of leading ANN models in decoding accuracy.
- RAT SNN is a lightweight and low computational cost model that achieves lower power consumption with fewer parameters and high decoding accuracy, and provides a promising solution for fully iBMIs.

## 2 RELATED WORKS

### 2.1 ANN-BASED CST DECODER

Numerous studies have employed ANNs for CST movement decoding (Ye & Pandarinath, 2021; Ye et al., 2023; Liu et al., 2022; Schneider et al., 2023; Le & Shlizerman, 2022; Pandarinath et al.,

2018). As an example for iBMIs, AEGRU (Wang et al., 2024) incorporates an autoencoder-based training module to boost performance while maintaining training efficiency and lightweight inference. Although achieving excellent performance within specific training experimental conditions, these works largely ignore the challenges posed by neural variability across subjects and tasks. To address neural drift across time, pioneering work has made significant efforts (Farshchian et al., 2019; Karpowicz et al., 2025; Dyer et al., 2017; Degenhart et al., 2020). NOMAD (Karpowicz et al., 2025) and ADAN (Farshchian et al., 2019) establishes baseline models on anchor sessions, then aligns subsequent sessions through dynamic time warping. While improving cross-session consistency, this alignment relies on the assumption of shared dynamics or task structure. To tackle broader neural variability, POYO (Azabou et al., 2023) and NDT3 (Ye et al., 2025) employ multi-layer transformer architectures trained on large-scale neural activity data, enhancing generalization through scaling up. NDT3 utilizes a simple tokenizer for neural firing rates, whereas POYO constructs UnitEmbed representations by tokenizing neural spiking sequences, enabling rapid finetuning across subjects. However, these previous methods rely on large models and significant computational resources, hindering deployment in resource-constrained fully iBMIs. In this work, we introduce a re-parametrization method to replace the tokenizer with a subject-specific BN layer, which can be seamlessly integrated into the linear operations between spiking neurons. This modification enables RAT SNNs to be pretrained on large datasets and then rapidly adapted to new subjects and tasks, while maintaining a lightweight architecture without additional computational overhead.

## 2.2 SNN-BASED CST DECODER

As representative low-power models, SNNs have been explored for iBMIs (Yik et al., 2025; Taeckens & Shah, 2024; Liao et al., 2022; Zhou et al., 2024; Liu et al., 2025; 2024). However, recent studies primarily emphasize power efficiency, often yielding suboptimal decoding accuracy compared to ANNs. Although the SNN3d model proposed by NeuroBench demonstrates significant gains over classical SNNs, its incorporation of MAC operations between synapses substantially increases computational complexity leading to higher power consumption, deviating from core SNN principles (Yik et al., 2025). Currently, bigRSNN stands as the best-performing SNN model reported for CST decoding by leveraging cross-session pretraining (Liu et al., 2024). However, these studies predominantly rely on simplistic SNN architectures, and have not effectively explored the potential of SNNs to serve as a foundation model capable of addressing neural variability across time, subject, and tasks. Furthermore, Spikachu (Mentzelopoulos et al., 2025) incorporates attention mechanisms but relies on an ANN harmonizer to address neural variability, introducing massive MAC operations. In contrast, we propose RAT SNN, a lightweight pure SNN architecture that integrates re-parameterized spike-driven self-attention with multi-timescale dynamics to achieve high decoding accuracy while maintaining AC-only operations for adhering to strict power constraints.

## 3 METHOD

To develop a compact yet effective neural decoder, we propose RAT SNN of only 4 layers of LIF neurons, including a CST re-parametrized attention block and a readout block (Fig. 1). RAT SNN incorporates 3 key components: 1) Dynamic synapse mechanisms with recurrent connections, which enhance the ability to capture multi-timescale neural dynamics; 2) CST re-parameterization self-attention, which improves training efficiency and final performance; 3) Subject-specific BN and a cross-condition training framework, which further enhance decoding performance and the generalization ability of RAT SNN-CC. In the remainder of this section, we will introduce detailed descriptions of each of these three components.

### 3.1 MULTI-TEMPORAL DYNAMIC SPIKING NEURON

The CST re-parametrized attention block in RAT SNN employs LIF neurons as fundamental computational units (Maass, 1997), with discrete-time dynamics defined as:

$$H[t] = \alpha V_{\text{mem}}[t-1] + V_{\text{syn}}[t-1] \tag{1}$$
$$S[t] = \text{Hea}(H[t] - V_{\text{th}}) \tag{2}$$
$$V_{\text{mem}}[t] = H[t](1 - S[t]) + V_{\text{reset}}S[t] \tag{3}$$

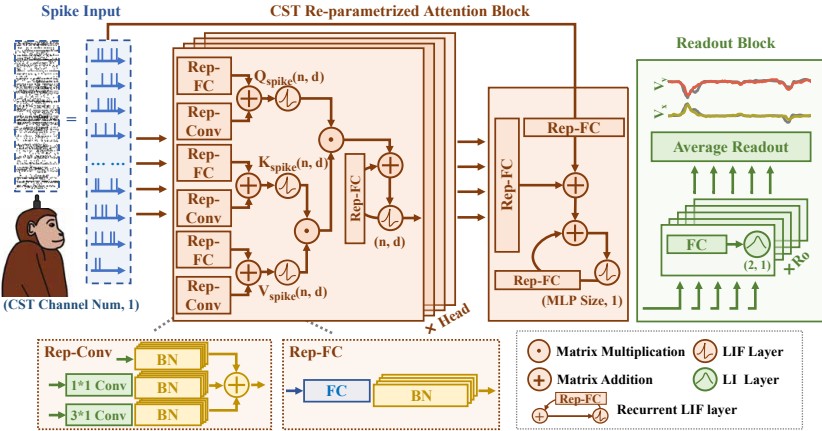

Figure 1: **RAT SNN overall architecture.** Each time step, RAT SNN takes spikes input with dimension $(\text{CST Channel Num}, 1)$ from cortical spike train (CST), and outputs kinematic predictions after processing through `CST re-parametrized attention block` and `readout block`. Rep-FC denotes a re-parameterized fully connected layer, `Rep-Conv` refers to a three-branch 1D convolutional layer. $n$ and $d$ represent the number of tokens and the feature dimension of self-attention, `MLP Size` indicating output dimension of CST re-parametrized attention block. `Head` and `Ro` indicates number of attention heads and multi-readout block.

where $H[t]$ denotes membrane potential before neuronal dynamics at time step $t$, $V_{\text{mem}}[t-1]$, $V_{\text{syn}}[t-1]$ are current membrane and synaptic potential, $\alpha < 1$ is decay coefficient. $\text{Hea}(\cdot)$ is the Heaviside step function, indicating that output spike $S[t]$ emits when $H[t]$ exceeds threshold $V_{\text{th}}$, which resets to $V_{\text{reset}}$ afterwards; otherwise, $H[t]$ continues to decay without spiking (Rossbroich et al., 2022).

To simulate the temporal dynamic adaptability of biological neuron synapses and enhance the multi-timescale dynamics, we consider the recurrently connected LIF neuron with synaptic dynamics as Recurrent-LIF (Fig. 2), and maintain heterogeneity among intralayer neurons (Perez-Nieves et al., 2021). Specifically,

$$V_{\text{syn}}^l[t] = \sum_{i=1} f_{op_i}^l\left(S_i^u[t]\right) + f_{\text{Rec}}^l(S^l[t-1])$$
$$+ \beta^l V_{\text{syn}}^l[t-1] \qquad (4)$$

where $f_{op}^l(\cdot)$ denotes the feedforward connections of upstream spike $S^u$ in LIF layer $l$ with $i$ indicate different branches. And $f_{\text{Rec}}(\cdot)$ denotes the recurrent connections, we use re-parameterization-style fully-connection (FC) here. $\beta < 1$ represents the synaptic decay factor. Recurrent FC forming internal layer-wise dynamics. Together with inter-layer connections, it mimics the coexistence of long-range projection pathways and local microcircuits in biological neural systems.

Figure 2: **Recurrent-LIF structure.** Recurrent-LIF neuron receives upstream spikes $S^u$ and intra-layer projection $f_{\text{Rec}}(S^l)$, updates $V_{\text{syn}}$ and $H$, and then emits spikes $S^l$ and updates $V_{\text{mem}}$ via $\text{Hea}(\cdot)$.

The readout block consists of five heads, each comprising two Leaky Integrator (LI) neurons to predict the two-dimensional forelimb movement velocity. LI neurons share the same membrane potential accumulation and leakage processes as LIF neurons without a spiking mechanism, aiming to simulate the smooth motion characteristics of muscle groups, thereby converting the discrete spiking activity $S[t]$ into continuous outputs:

$$H^{\text{LI}}[t] = V_{mem}^{\text{LI}}[t] = \alpha^{\text{LI}} V_{mem}^{\text{LI}}[t-1] + V_{syn}^{\text{LI}}[t-1] \qquad (5)$$

### 3.2 CST RE-PARAMETRIZED SELF-ATTENTION

The main body of the CST attention block is a multi-head self-attention. By leveraging the binary nature of SNN, the computational complexity of the attention operation can be reduced from $O(n^2 d)$

to $O(nd^2)$, as in linear attention, where $n \gg d$. In combination with Recurrent-LIF:

$$\text{Attention}_{s_i} = \text{RLIF}((Q_{s_i} K_{s_i}^T) V_{s_i}) = \text{RLIF}(Q_{s_i} (K_{s_i}^T V_{s_i})) \tag{6}$$

where $Q_{s_i}$, $K_{s_i}$ and $V_{s_i}$ respectively represent query(Q), key(K) and value(V) of $i$-th attention head with spike representation, and $\text{RLIF}(\cdot)$ denotes the temporal dynamics of Recurrent-LIF.

Previous studies have shown that convolution layer (Conv) can effectively enhance the performance of spike-driven attentions (Yao et al., 2023b). However, considering the discontinuity of electrode channel indices and inter-subject differences, we integrate parallel Conv and FC for Qs/Ks/Vs, where FC captures global features and Conv focuses on local features across CST channels:

$$Q_s / K_s / V_s = \text{RLIF}(\text{Rep-Conv}(X_{CST}) + \text{Rep-FC}(X_{CST})) \tag{7}$$

where $X_{CST}$ denotes the input CST spikes at current time step and $\text{Rep-Conv}(\cdot)$ and $\text{Rep-FC}(\cdot)$ denote the re-parameterized Conv and FC.

Moreover, previous experiments have shown that overly deep SNNs can impair decoding accuracy and real-time efficiency in CST decoding for BCI systems and other applications (Rossbroich et al., 2022). Therefore, we merge the MLP block and the output layer in the attention block from prior spike-driven transformer designs (Yao et al., 2023a;b; Zhou et al., 2023; 2022), reducing the depth from the classic five layers to three; this shallower structure not only helps improve accuracy but also reduces model size and computational operations.

To further reduce model depth, we compute Qs/Ks/Vs directly from the input CST spikes and, since they lack membrane potentials, replace the membrane-shortcut (MS) in spike-driven transformers with a spike-element-wise shortcut (Fang et al., 2021) followed by a linear layer (a "surrogate short-cut"):

$$U_s = \text{RLIF}(\text{Attention}_s + \text{Rep-FC}(X_{CST})) \tag{8}$$

where $U_s$ denotes the output spikes of CST re-parametrized attention block at current time step. Surrogate shortcut enables changing the attention output dimension (MLP Size $> n \times d$, Fig. 1), providing flexible output dimensionality and improving performance without increasing depth.

### 3.2.1 RE-PARAMETRIZED SELF-ATTENTION

Furthermore, structural re-parameterization was originally proposed to address issues such as gradient vanishing and suboptimal performance in VGG (Ding et al., 2021). However, it has also been found to be highly beneficial for SNNs (Yao et al., 2023a), enabling the fusion of complex inter-layer structures in SNNs into a single linear connection, thereby promoting training stability and keeping the pure AC operations (without MAC operations) between LIF neurons during the inference phase.

Building on this, we extend the CST attention block to the CST re-parametrized attention block that leverages BN during training to improve performance and accelerate convergence. Specifically, we introduce two re-parameterized structures: Rep-Conv (1D Conv) and Rep-FC.

Inspired by RepConv2d in RepVGG (Ding et al., 2021), Rep-Conv employs multiple parallel branches, including $1 \times 1$ kernel $\text{Conv}_1(\cdot)$, $3 \times 1$ kernel $\text{Conv}_3(\cdot)$, optionally larger kernels $\text{Conv}_i(\cdot)$ ($i = 5, 7, \ldots$) and identity mapping $\text{id}(\cdot)$ which can be viewed as Conv operation with an identity kernel $\text{Conv}_0(\cdot)$. Given an input $X_c$:

$$\text{Rep-Conv}(X_c) = \sum_{i \in \mathcal{K}} \text{Conv}_i(X_c) = \sum_{i \in \mathcal{K}} \text{BN}_i(W_i * X_c + b_i) \tag{9}$$

where $*$ denotes the Conv operation, $W_i$, $b_i$ and $\text{BN}(\cdot)$ denotes kernel, bias and BN of the $i$-th branch, $\mathcal{K} = \{0, 1, 3, \ldots\}$. The output of Rep-Conv is a weighted sum of outputs from all branches, which can be re-parameterized into a single Conv during inference:

$$\text{Rep-Conv}(X_c) = (\sum_{i \in \mathcal{K}} \text{PAD}(W_i^{\text{Rep}})) * X_c + \sum_{i \in \mathcal{K}} b_i^{\text{Rep}} \tag{10}$$

where $W_i^{\text{Rep}} = \frac{W_i}{\sqrt{\sigma_i^2 + \epsilon}} \gamma_i$, $b_i^{\text{Rep}} = \frac{b_i - \mu_i}{\sqrt{\sigma_i^2 + \epsilon}} \gamma_i + \beta_i$ denote re-parameterized Conv kernel and bias of branch $i$, while $\mu_i$, $\sigma_i$, $\gamma_i$, $\beta_i$, and $\epsilon$ are parameters of corresponding BN. And $\text{PAD}(\cdot)$ means padding the Conv kernel of all branches to the largest one. Similarly, with weight matrix $W$, given an input $X_f$, Rep-FC can be expressed as:

$$\text{Rep-FC}(X_f) = \text{BN}(W X_f) = W_i^{\text{Rep}} X_f \tag{11}$$

## 3.3 STEPWISE TRAINING FRAMEWORK

In this work, we encapsulated a composite dataset composed of multiple subjects and tasks from different sources. Each subject contributes to one or more sessions, where each session represents data collected during a distinct time period. The data were split into training, validation, and testing sets at the session level. Depending on the scale of the training set, this dataset was organized into three hierarchical levels: cross-condition (encompassing multiple subjects and tasks), cross-session (same as in-subject), and single-session.

Due to the variability of neural activity, there exist various distributional differences across subjects, tasks and sessions. As a result, even a well-trained cross-condition model may struggle to generalize effectively when transferred directly to a specific session. To better leverage the potential of data at different scales and enable the model to gradually adapt from the broader cross-condition distribution to the more specific session-level distribution, we divide the training pipeline of RAT SNN-CC into four distinct stages based on dataset granularity: 1) cross-condition pretraining, 2) cross-session retraining, 3) single-session finetuning, and 4) optional lightweight finetuning (Fig. 3).

Notice that, benefiting from the reparameterization, we assign an independent BN for each condition during the cross-condition pretraining phase (Fig 3, top). During each epoch, BN are switched dynamically according to the subject identity of the current batch, enabling RAT SNN-CC to better accommodate the distributional drifts across different conditions. For datasets with only one session or generalization experiments, we skip the cross-session retraining step and directly perform finetuning on the target session. In the lightweight finetuning stage, we first finetune the model with an activity upper bound (AUB) of each individual LIF neuron after structural re-parameterization to reduce the overall firing rate. Then, we provide an optional iterative prune step (masked the smallest $p$ of weights) and following retrain to further reduce model connection size and computational complexity.

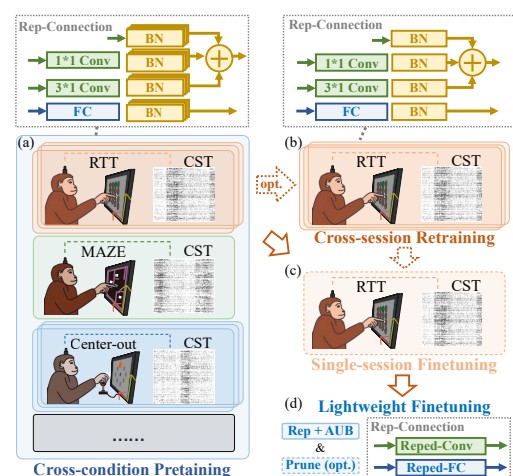

Figure 3: **Training pipeline.** (a) Cross-condition: training across different experimental conditions, with a specific BN assigned to each subject. (b) Cross-session: training on multi-session data from the target condition, with BN fixed to the corresponding sets. (c) Finetune on the target session. (d) Further optimize via structural reparameterization and pruning.

## 4 EXPERIMENTS

### 4.1 DATASETS AND EXPERIMENT SETUP

A key advantage of Pretained-RAT SNN lies in ability to leverage the discrete spatiotemporal processing capabilities of SNNs with pretraining strategies. This enables effective integration of neural variability across conditions, including across time, subjects, tasks, experimental environments. To this end, we integrate data from multiple datasets (Table 1). All these data are derived from electrophysiological recordings of the M1, PMd, and S1 regions in monkeys, spanning 103 sessions across six subjects. Neural activity in these regions has been extensively studied, deep learning tools and benchmarks have already been established as well (Pei et al., 2021).

These neural recordings correspond to movements of subjects while performing three different tasks: random target task (RTT), maze target reach (MAZE), and center-out(CO). Specifically, RTT requires subjects to continuously and self-pacedly move a cursor on the screen via finger motion (or joystick in RTT-joystick) to reach randomly appearing targets. In MAZE, the subject has to move the cursor around obstacles to reach a designated target within each trial. And CO requires the subject to use a joystick to control the cursor from screen center to peripheral targets per trial. Data from MAZE and CO were trial-segmented due to their discrete trial structure.

Table 1: Datasets used in this work. "*" means being used in cross-condition dataset, while "+" means being used in cross-session dataset. RTT: Random Target Task, CO: Center-Out.

| Study | Subject | Brain Area | Channels | Task | Sessions |
|---|---|---|---|---|---|
| NHP (O'Doherty et al., 2017)[*+] | Monkey I, L | M1, S1 | 96/192 | RTT | 37 |
| NLB_MAZE (Pei et al., 2021)[*] | Monkey Je | M1, PMd | 192 | MAZE | 1 |
| NLB_RTT (Pei et al., 2021) | Monkey I | M1 | 96 | RTT | 1 |
| Perich dataset (Dyer et al., 2017) | Monkey M, J, C, T | M1, PMd | 192 | RTT-joystick, CO | 53 |
| Private dataset (Liu et al., 2025)[*+] | Monkey C05 | M1 | 96 | CO | 11 |

Table 2: Performance ($R^2 \times 100$) on NHP dataset (O'Doherty et al., 2017). "-SS": single-session from scratch; "-CS": cross-session pretrained on same subject; "-CC": cross-condition pretrained across sessions, subjects, and tasks. Results without standard deviation (std) are from published works. Best and second-best models are in bold and underlined, respectively. "-P" indicates pruned.

| Model | | Monkey I | Monkey L | Average |
|---|---|---|---|---|
| Classical | KalmanFilter (Dangi et al., 2011) | 21.13 | 14.20 | 17.67 |
| ANN | ANN2d (Yik et al., 2025) | 59.28 | 55.82 | 57.55 |
| | ANN3d (Yik et al., 2025) | 64.27 | 58.90 | 61.60 |
| | GRU (Wang et al., 2024) | 71.00 | 62.33 | 67.00 |
| | AEGRU (Wang et al., 2024) | 72.00 | 67.00 | 69.50 |
| | POYO-SS (Azabou et al., 2023) | 62.33±0.85 | 60.53±2.32 | 61.43±1.25 |
| | POYO-CS (Azabou et al., 2023) | 70.99±0.42 | 69.63±2.32 | 70.31±0.18 |
| SNN | SNN2d (Yik et al., 2025) | 59.30 | 55.83 | 57.57 |
| | SNN3d (Yik et al., 2025) | 64.20 | 62.40 | 63.30 |
| | bigRSNN-SS (Liu et al., 2024) | 70.46±0.24 | 65.40±0.88 | 67.94±0.37 |
| | bigRSNN-CS (Liu et al., 2024) | 70.89±0.45 | 68.70±0.36 | 69.79±0.22 |
| Ours | RAT SNN-SS | 72.22±0.38 | 66.30±0.95 | 69.26±0.56 |
| | RAT SNN-CS | **74.26±0.14** | 68.63±0.44 | 71.45±0.15 |
| | RAT SNN-CC | 74.06±0.42 | 70.40±0.68 | **72.23±0.38** |
| | RAT SNN-CC-P | 73.18±0.17 | **70.75±1.24** | 71.96±0.61 |

We constructed the cross-condition dataset with Monkey I, Monkey L in NHP (O'Doherty et al., 2017), Monkey Je in NLB (Pei et al., 2021), and Monkey C05 in private dataset (Liu et al., 2025) for Cross-condition RAT SNN (-CC). Cross-session RAT SNN (-CS) was pretrained across all available sessions from Monkey I, Monkey L, and Monkey C05 respectively, followed by finetuning on individual sessions. In contrast, single-session RAT SNN (-SS) was trained directly on the target session. The test segments were selected from three representative sessions of Monkey I and Monkey L as recommended by NeuroBench (Yik et al., 2025), and from randomly selected sessions in the remaining datasets.

## 4.2 TESTING ON SINGLE-SESSION

We first evaluated the performance of RAT SNN at single-session level on Monkey I and Monkey L (Table2). Results show that RAT SNN-SS achieves decoding accuracy comparable to SOTA SNN (cross-session pretrained bigRSNN) and ANN (POYO) models, while also maintaining competitive performance under other conditions (Table3, 4). Notably, RAT SNN delivers these results with 600K parameters, significantly fewer than bigRSNN (1.2M) and POYO-SS (1.9M).

Table 3: Performance ($R^2 \times 100$) on Monkey C05. "+" means generalize result on unseen session.

| | Model | C05_2022 | C05_2025 |
|---|---|---|---|
| ANN/SNN | AEGRU | 76.05±0.43 | 52.39±0.81 |
| | POYO-SS | 77.05±0.80 | 64.93±2.12 |
| | bigRSNN-SS | 81.72±0.35 | 62.15±0.63 |
| | bigRSNN-CS | 83.56±0.54 | 66.45±0.28[+] |
| Ours | RAT SNN-SS | 81.58±0.42 | 66.71±0.48 |
| | RAT SNN-CS | 84.34±0.19 | 66.77±1.11[+] |
| | RAT SNN-CC | **85.07±0.21** | 66.54±0.74[+] |
| | RAT SNN-CC-P | 83.88±0.11 | **67.17±1.18[+]** |

Table 4: Performance ($R^2 \times 100$) on NLB dataset (Yik et al., 2025). "$+$" means generalize result on unseen session. "—": not applicable.

| Model | | NLB_MAZE
Monkey Je | NLB_RTT
Monkey I |
|---|---|---|---|
| ANN | GRU (Azabou et al., 2023) | 88.87 | 59.51 |
| | AutoLFADS (Pandarinath et al., 2018) | 90.62 | 59.31 |
| | NDT (Ye & Pandarinath, 2021) | 89.29 | 58.95 |
| | NDT-Sup (Liu et al., 2022) | 87.08 | 46.21 |
| | EIT (Liu et al., 2022) | 87.91 | 46.91 |
| | POYO-SS (Azabou et al., 2023) | 94.70 | 68.50 |
| | POYO-1 (Azabou et al., 2023) | **94.82**$^+$ | 73.78$^+$ |
| SNN | bigRSNN-SS (Liu et al., 2024) | 90.38±0.06 | 71.66±0.54 |
| | bigRSNN-CS (Liu et al., 2024) | — | 78.61±0.52$^+$ |
| Ours | RAT SNN-SS | 90.76±0.30 | 76.34±0.54 |
| | RAT SNN-CS | — | 78.55±0.19$^+$ |
| | RAT SNN-CC | 91.16±0.19 | **78.70±0.66**$^+$ |
| | RAT SNN-CC-P | 91.02±0.15 | 78.08±0.75$^+$ |

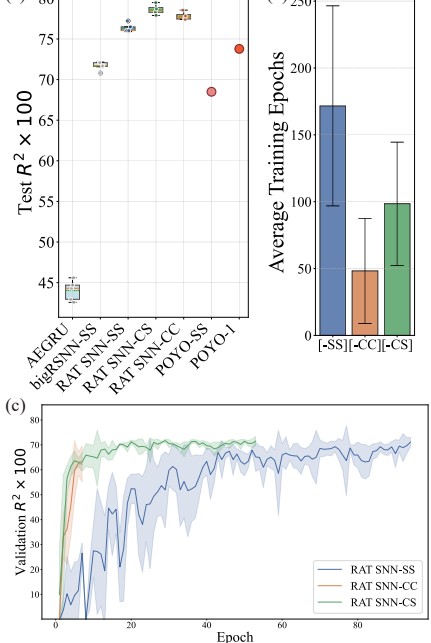

Figure 4: **Result of neural temporal drift generalization on NLB_RTT.** (a) $R^2$ compared to baselines. (b) Training epochs comparison. (c) Learning curves, shaded area denotes std, aligned to fastest converging seed.

Figure 5: **Generalization to unseen conditions.** (a) Average training epochs of RAT SNN-SS and RAT SNN-CC on unseen subject sessions and unseen subject with unseen task (RTT-joystick) sessions. (b-c) Learning curves on example sessions.

### 4.3 GENERALIZING ACROSS NEURAL TEMPORAL DRIFT

Furthermore, we evaluated the performance of RAT SNN-CS on seen sessions and its generalization capability to unseen sessions. On seen sessions, cross-session pretraining effectively improved the decoding accuracy of RAT SNN, enabling it to surpass single-session version and baselines in most sessions (Table 2, 3). Although this is not a standard generalization experiment, exploring ways to enhance decoding performance on seen sessions can provide valuable insights for neuroscience research based on AI—AI for neuroscience (Zador et al., 2023; Chen et al., 2022).

To evaluate temporal generalization of RAT SNN-CS, we use the NLB_RTT dataset (Pei et al., 2021) (same animal and task as NHP, Monkey I). Remarkably, although RAT SNN-SS already achieved excellent performance on NLB_RTT, even substantially surpassing POYO-1 (pretrained on cross-condition, 11.8M), RAT SNN-CS further improved performance (Table 4). The learning curves further demonstrate that RAT SNN-CS converges faster than its single-session counterpart (Fig. 4), proving that RAT SNN-CS can rapidly generalize to unseen sessions from different time periods, demonstrating excellent capability of addressing neural temporal drift.

### 4.4 GENERALIZING ACROSS NEURAL VARIABILITY BETWEEN SUBJECTS AND TASKS

After demonstrating that RAT SNN possesses strong capabilities in integrating cross-session data and generalizing to neural temporal drift, we further validate its ability to integrate across conditions. After cross-condition pretraining, RAT SNN-CC achieved superior performance on both seen and unseen sessions of seen subject and seen conditions, highlighting the effectiveness of cross-condition pretraining and its capacity to integrate neural variability (Table 2, 3, 4). Notably, compared to POYO, RAT SNN-CC consistently outperformed its single-session version across multiple datasets, suggesting more substantial performance gains and superior integration of heterogeneous data.

Furthermore, we evaluated the generalization capability of RAT SNN-CC to unseen conditions (without cross-session re-training). Result show that even when generalizing to an entirely unseen subject performing an unseen task (RTT-joystick), the RAT SNN-CC achieves faster convergence and superior performance compared to the version trained from scratch, highlighting its strong cross-condition generalization (Fig. 5).

### 4.5 STRUCTURAL ABLATION STUDY

Using the small-scale dataset (Monkey C05) as an example, we conduct ablation experiments on the RAT SNN-SS to evaluate how CST re-parameterized attention block affects decoding accuracy.

Comparing our CST attention block (MLP size= 512) with the classical spike-driven transformer block (SDT SNN) and RAT SNN-192 (MLP size= 192, the same output dimension as the SDT block and equal to the input size), our method yields higher decoding accuracy, and enlarging its output dimension via the "surrogate shortcut" further improves performance (Table 5).

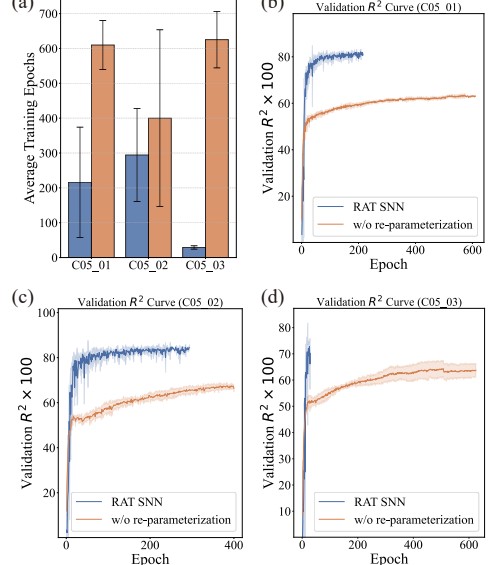

Table 5: Ablation study results on RAT SNN-SS on Monkey C05.

| Model | C05_2022 | C05_2025 |
|---|---|---|
| SDT SNN | 80.09±0.23 | 65.21±0.57 |
| RAT SNN-192 | 80.49±0.38 | 66.61±0.79 |
| RepFC SNN-H1 | 81.09±0.40 | 66.26±0.35 |
| RepFC SNN-H3 | 79.47±0.96 | 65.74±1.00 |
| **RAT SNN-SS** | **81.58±0.42** | **66.71±0.48** |

Table 6: Ablation study results on re-parameterization on Monkey C05.

| Model | $R^2 \times 100$ | Epoch |
|---|---|---|
| w/o re-param | 67.58±2.50 | 545±84 |
| **RAT SNN-SS** | **81.58±0.42** | **179±67** |

Figure 6: **Performance comparison between RAT SNN and w/o re-parameterization version.** (a) Average training epochs for both models on Monkey C05. (b-d) $R^2$ learning curves.

Also, compared with fully connected recurrent SNN baselines (RepFC SNN), our method outperforms both single-hidden-layer (RepFC SNN-H1) and 3-hidden-layer (-H3) variants (Table 5), showing the effectiveness of CST attention block in capturing spatial dependencies in CST.

Furthermore, we investigate the impact of the re-parameterization method on RAT SNN (Table 6). This strategy both accelerates learning and improves decoding accuracy: relative to the standard RAT SNN, the w/o re-parameterization variant shows a 17.16% drop in $R^2$ and requires roughly three times more training epochs to converge (Fig. 6).

## 4.6 REDUCING COMPUTATIONAL CONSUMPTION WITH HIGH PERFORMANCE

To evaluate computational cost of RAT SNN, we used NeuroBench (Yik et al., 2025) to count ACs and MACs on NHP dataset (O'Doherty et al., 2017) as energy metrics.

With the re-parameterization method, RAT SNN involves only synaptic AC operations during inference, aligning with the spike-driven nature of energy-efficient SNNs (Table 7). Furthermore, with simple pruning strategy, RAT SNN-CC-P substantially reduces computational cost while maintaining comparable performance (Wilcoxon test, p=0.3828, Table 2). Compared to basic SNN model (SNN2d), although incurring higher ACs, RAT SNN-CC-P still remains energy efficient given the substantial performance gap. Compared to bigRSNN, RAT SNN-CC-P has a substantially lower computational cost and a smaller parameter size (150K vs. 1.2M), making it more suitable for deployment in fully implantable iBMIs.

Table 7: Comparison of synaptic operations across different Models.

| Model | Synaptic Operations | |
|---|---|---|
| | Effective MACs | Effective ACs |
| ANN2d | 4970 | 0 |
| ANN3d | 11507 | 0 |
| AEGRU | 25095±69 | 0 |
| POYO | 1730507±3 | 810339 |
| SNN2d | 0 | 414 |
| SNN3d | 32256 | 5831 |
| bigRSNN | 0 | 42003±6281 |
| RAT SNN-CC | 0 | 65307±841 |
| RAT SNN-CC-P | 0 | 21020±5515 |

Furthermore, RAT SNN is significantly more efficient than other operation-efficient ANN-based decoders (AE-GRU and basic ANN models), especially considering that 1 MAC operation consumes approximately 31 times the energy of an AC operation (Yin et al., 2021). Therefore, RAT SNN demonstrates an outstanding performance-to-energy ratio, making it a highly promising candidate for future fully iBMI applications.

## 5 CONCLUSION

In this work, we propose RAT SNN, a CST decoder that integrates re-parameterized self-attention mechanisms. RAT SNN achieves superior decoding performance through cross-condition pretraining and possesses strong generalization capabilities, enabling rapid finetuning on unseen data to overcome neural temporal drift in individual subjects as well as neural variability across other cross-condition scenarios. Meanwhile, RAT SNN features a lightweight architecture and strictly adheres to the spike-driven nature of low-power SNNs, realizing a prototype SNN foundation model with high performance, strong generalization, and low energy consumption (Fig.7).

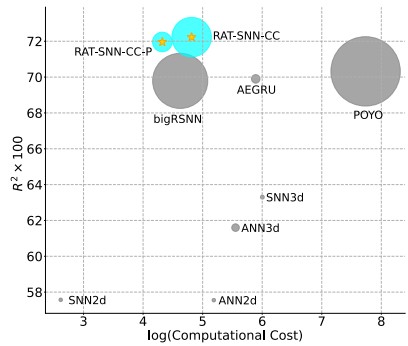

Figure 7: **Comparison summary between RAT SNN and baselines.** Computational Cost = Effective ACs + Effective MACs × 31 (Yin et al., 2021). Bubble size represents the model size.

ACKNOWLEDGMENTS

This work is supported by: the STI 2030-Major Projects (No.2022ZD0208604), the Key R&D Program for Zhejiang (No. 2021C03003, 2022R52033, 2023C03081, 2023C03026, 2024C03002), and the Lingang Laboratory (Grant No.LGL-1987-08).

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

## APPENDIX

## A DATASETS

We conducted experiments on five datasets, involving various subjects and tasks, as summarized in Table 1. Based on these datasets, we established a cross-condition dataset incorporating data from multiple conditions, including different subjects, tasks, and sessions. Notably, the subject Monkey I in NHP dataset (O'Doherty et al., 2017) was included in NLB dataset (Pei et al., 2021), but with different sessions. Private dataset Liu et al. (2025) includes 10 sessions of Monkey C05 recorded in 2022 (C05_2022), and 2 sessions recorded in 2025 (C05_2025). The RTT task in Perich dataset (Perich et al., 2018) is different from other dataset, with subjects could not touch the screen to move cursor but with joystick.

We evaluated RAT SNN with the standardized brain-inspired benchmarking framework, NeuroBench (Yik et al., 2025). All samples from the datasets were uniformly downsampled at 4 ms intervals, with z-score normalization applied per-session basis. The model was trained and validated using overlapping 2-second samples, while testing employed continuous long sequences to simulate real-time processing. For each session, 10% of the data was reserved for validation and 25% for testing following NeuroBench recommendations (Fig. 8) (Yik et al., 2025), except for the Perich dataset. For Perich dataset, we reserved 10% of the data for validation and 10% for testing as recommended by POYO (Azabou et al., 2023).

### A.1 DATASETS FOR MULTIPLE TRAINING STAGES

We constructed three hierarchical datasets corresponding to four training stages of RAT SNN (Fig. 9). 1) The cross-condition dataset is used in the cross-condition pretraining stage of cross-

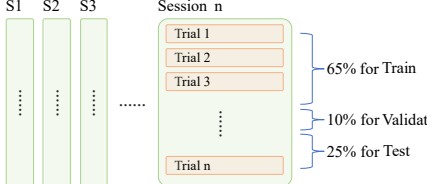

Figure 8: **Dataset segmentation schematic.** Each session consists of multiple trials. We divide the data into training, validation, and test sets based on trials. Using NHP dataset (O'Doherty et al., 2017) as a segmentation example.

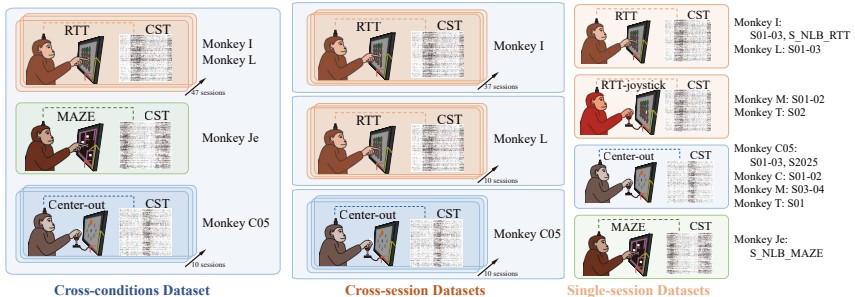

Figure 9: **Datasets for Multiple Training Stages.** We constructed three datasets corresponding to the cross-condition, cross-session, and single-session levels. Left: The cross-condition dataset includes data from different subjects and tasks. Middle: The cross-session dataset includes multiple sessions from the same subject. Right: The single-session dataset includes a single session from a single subject, specific session indices are annotated on right. Different subject monkeys are represented by different colors, and overlapping boxes with arrows indicate multiple sessions from the same subject.

condition RAT SNN (RAT SNN-CC), and includes data from different subjects and tasks. 2) The cross-session dataset is used in both the cross-session pretraining stage of cross-session RAT SNN (RAT SNN-CS) and the cross-session retraining stage of RAT SNN-CC. It consists of multiple sessions from the same subject. 3) The single-session dataset is used in the training stage of the single-session RAT SNN (RAT SNN-SS), as well as in the single-session finetuning and lightweight finetuning stages of both the RAT SNN-CC and RAT SNN-CS. It contains data from a single session of a single subject.

The cross-condition dataset comprises data from different subjects and tasks, including Monkey I and Monkey L from NHP dataset (O'Doherty et al., 2017), Monkey Je from NLB_MAZE (Pei et al., 2021), and Monkey C05_2022 from our private dataset (Liu et al., 2025). Cross-condition pretraining was performed using this combined dataset to train the RAT SNN-CC, which was subsequently finetuned on individual downstream subjects and tasks.

The cross-session datasets includes multiple sessions from the same subject. In this study, we used all available sessions from Monkey I, Monkey L (NHP dataset), and Monkey C05_2022 (private dataset) to train and evaluate the RAT SNN-CS.

The single-session dataset contains only one session from a single subject. For NHP dataset, we selected three representative sessions from Monkey I and Monkey L, as recommended by NeuroBench (indy_20160622_01, indy_20160630_01, indy_20170131_02, loco_20170210_03, loco_20170215_02, loco_20170301_05 Yik et al. (2025)).

For NLB_MAZE and NLB_RTT (Pei et al., 2021), we used the only available sessions from Monkey Je ("MAZE_000128") and Monkey I ("RTT_000129"), respectively.

For Perich dataset (Perich et al., 2018) and the private dataset (Liu et al., 2025), we randomly selected early sessions from Monkey C05 (c05_20220120094153, c05_20220114093926, c05_20220114132255 and c05_20250221141323, c05_20250218095128), Monkey M (m_20140114_random_target_reaching, m_20140115_random_target_reaching, m_20140203_center_out_reaching, m_20140217_center_out_reaching), Monkey T (t_20130819_center_out_reaching, t_20130820_random_target_reaching), and Monkey C (c_20161011_center_out_reaching, c_20161013_center_out_reaching), prioritizing temporally consecutive recordings to ensure result stability.

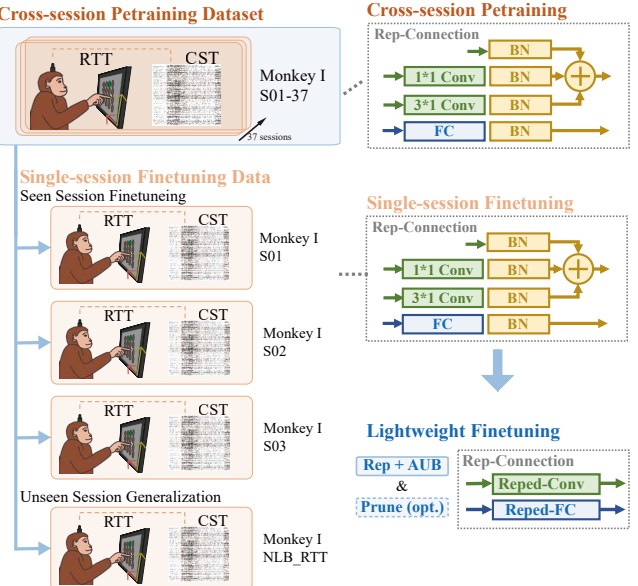

Figure 10: **Cross-session training pipeline.** Using Monkey I as an example: RAT SNN was first pretrained on the cross-session dataset (from NHP dataset), then separately finetuned on three sessions from NHP dataset to evaluate decoding performance on seen sessions, and the extra session from NLB_RTT to assess generalization capability on unseen sessions.

# B  TRAINING

## B.1  CROSS-SESSION PRETRAINING FOR NEURAL TEMPORAL DRIFT GENERALIZATION

Since a subject's dataset typically contains multiple sessions collected on different dates, with some sessions separated by intervals of even years, neural activity continuously changes over time–Neural Temporal Drift (Driscoll et al., 2022). Consequently, cross-session neural variability represents one of the primary challenges in iBMI decoding.

In this work, We enhance RAT SNN's neural temporal drift generalization performance through cross-session pretraining. And we further divide the neural temporal drift generalization performance evaluation into two components: generalization capabilities on seen sessions and unseen sessions. The generalization capability on seen sessions indicates that the model can cover neural temporal drift in observed data during training and extract underlying features during dynamic changes. The generalization capability on unseen sessions indicates that the model can rapidly adapt to feature changes in unseen sessions based on underlying features extracted from seen sessions, thereby simulating long-term deployment scenarios in real iBMI applications.

Specifically, we pretrained RAT SNN on the cross-session datasets, which includes all available sessions from Monkey I, Monkey L, and Monkey C05 respectively. Pretraining and finetuning pipeline is shown in Fig. 10. The pretraining stage was conducted on the cross-session datasets respectively, followed by finetuning on individual sessions of the same subject to evaluate decoding performance on seen sessions. Additionally, we finetuned the model on the NLB_RTT and C05_2025 to assess its generalization capability across temporal intervals on unseen sessions. After cross-session pretraining, the finetuning process comprises two distinct phases: 1) Single-session finetuning: the model architecture remains unchanged as we directly adapt RAT SNN to target session characteristics through conventional finetuning. 2) Lightweight finetuning with pruning (Fig. 10 Right, Section B.3).

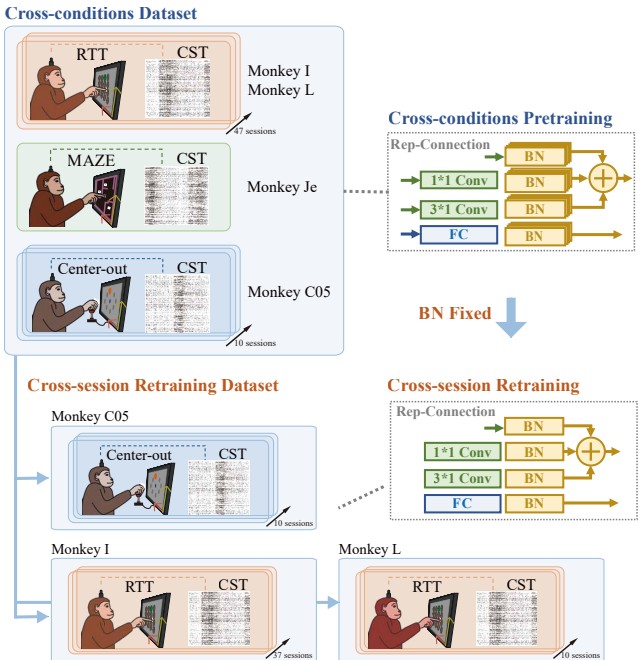

Figure 11: **Cross-condition training pipeline on subject with multiple session.** RAT SNN was first pretrained on the cross-condition dataset, which includes data from different subjects and tasks from Monkey I, Monkey L, Je, and C05. For Monkey I, L and C05, which have multiple sessions, RAT SNN-CC was then retrained on all training sessions of the same subject, and then finetuned on individual sessions with pipeline shown in the Fig.10.

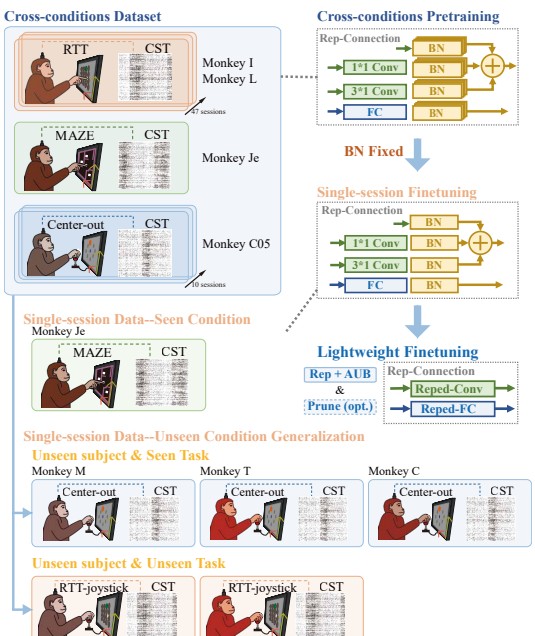

Figure 12: **Cross-condition training pipeline on subject with single session.** RAT SNN was first pretrained on the cross-condition dataset. For seen Monkey Je and unseen Monkey M, T, C, RAT SNN-CC was directly finetuned on the single-session data.

### B.2 Cross-condition pretraining for neural variability generalization across subjects and tasks

Foundation models for iBMI application need to possess two important characteristics: the ability to leverage cross-condition data to improve model decoding accuracy, and the ability to rapidly adapt to different subjects and tasks. Therefore, decoder models must effectively handle neural variability arising from different conditions including different subjects and tasks. When generalize to specific individuals, due to differences in experimental data richness across subjects, some subjects contain abundant data, allowing models to perform cross-session retraining on sufficient multiple session data to improve decoding accuracy. Other subjects only contain single-session data, so models must also possess the capability to directly finetune based on a limited amount of data to adapt to specific individual characteristics.

Based on above, we designed two different training strategies for RAT SNN-CC: 1) For subjects containing multi-session data, RAT SNN-CC undergoes pretraining on the cross-condition dataset, followed by retraining on all training sessions, and then finetuning on the target session (Fig.11), representing the highest decoding accuracy after fully utilizing all available data; 2) For subjects containing only single-session data or unseen condition generalization task, RAT SNN-CC is directly finetuned on the target session (Fig.12), encompasses both seen condition and unseen condition scenarios (including unseen subject with seen task and unseen subject with unseen task). The former represents ability to leverage large amounts of heterogeneous neural activity to improve decoding accuracy with limited samples, while the latter represents ability to rapidly adapt and improve decoding accuracy when facing unseen conditions, which is closer to real iBMI generalization scenarios. It is worth noting that for unseen conditions, we first average the parameters of all seen condition BN layers to initialize the BN parameters for the unseen condition, ensuring that the model can rapidly adapt to the features of unseen conditions.

### B.3 Lightweight finetuning and pruning

To reduce the energy consumption of RAT SNN and further stabilize the parameter space after re-parameterization, we merge the synaptic operation layers using the re-parameterization method after single-session finetuning, and then perform additional finetuning with an activity upper bound (AUB) regularization term. This enables the model to further adjust within the local parameter space, removing unnecessary active connections introduced by pretraining, and thus better adapting to the characteristics of the target session (Fig. 10 Right).

Additionally, we provide an optional pruning step to further reduce the computational cost of RAT SNN. Specifically, we setting the lowest prune percentage $p$ of weights to zero, and then perform masked retraining after pruning to recover performance. The detailed pruning procedure is shown in Algorithm 1. In this way, we can effectively reduce the computational cost of the model without significantly affecting its performance.

### B.4 Propagate style

Our method is built upon Stork (Zenke & Ganguli, 2018), which adopts a layer-connection propagation sequence inspired by the simulation of biological neural systems. Specifically, during the forward pass, Stork first updates the dynamics of all neural layers, followed by the update of all connections. This design implies that input at time step t only begins to influence the first neural layer at time $t + 1$, and the second layer at $t + 2$ (Fig. 13 Left).

While such a strategy may be advantageous in other domains with the biological plausibility, it introduces increasing latency between input and output in time series predict tasks which accumulates with model depth. To mitigate this issue, we employ sequential spike propagation scheme (Fig. 13 Right), where the dynamics of each neural layer and connections are updated sequentially within each time step, enabling the input at time step $t$ to influence the output within the same time step and allowing RAT SNN to decrease the latency between input and output.

---

**Algorithm 1** Iterative pruning

---

**Input:** RAT SNN parameter $\theta$, $R^2$ of validation before pruned $R_0^2$ ,start prune percentage $p$, performance decrease tolerance $\epsilon$, max prune percentage $p_{max}$, prune precision list $ps[p_1, p_2 \dots]$ (where $p_1 > p_2 > \dots$)
**Output:** Pruned RAT SNN parameter $\theta_{pruned}$
 1: precision idex $idx_p \leftarrow 0$, $R^2\_last \leftarrow 0$, $\theta_{pruned} \leftarrow False$
 2: **while** $p \leq p_{max}$ **do**
 3:  $R^2\_val, \theta' = $ Prune&Retrain(prune percentage $\leftarrow p$, model parameter $\leftarrow \theta$)
 4:  **if** $R^2\_val - R_0^2 < -|\epsilon \cdot R_0^2|$ **then**
      {if performance drop exceeds tolerance}
 5:   **if** $idx_p <$ len($ps$) **then**
 6:    $idx_p \leftarrow idx_p + 1$, $p \leftarrow p - ps[idx_p]$, **continue** {try smaller prune percentage}
 7:   **else if** $\theta_{pruned} == False$ **then**
 8:    $p \leftarrow p + ps[idx_p]$, **continue continue** {if not get a pruned model yet, continue to try}
 9:   **else**
10:    **break continue** {if already get a pruned model with minimum percentage, stop}
11:   **end if**
12:  **else**
13:   **if** $R^2\_val > R^2\_last$ **then**
14:    $R^2\_last \leftarrow R^2\_val$, $\theta_{pruned} \leftarrow \theta'$ {if performance improve, save}
15:   **end if**
16:   $p \leftarrow p + ps[idx_p]$ {update prune percentage}
17:  **end if**
18: **end while**
19: **if** $\theta_{pruned} == False$ **then**
20:  $\theta_{pruned} \leftarrow \theta$ {if never get a pruned model, return original model}
21: **end if**
22: **return** $\theta_{pruned}$

---

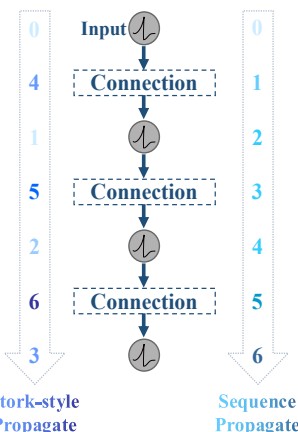

Figure 13: **Comparison between two propagation styles**. Left: Stork-style propagation, Right: sequential propagation. The internal propagation order within each time step is indicated by ascending numbers.

### B.5 BATCH NORMALIZATION IN RAT SNN

We apply two BN strategies in the CST re-parameterized attention block. First, for the Qs/Ks/Vs computation we use head-wise BN (i.e., a separate BN for each attention head) to preserve each head's independence before the attention calculation. Second, after the attention operation we apply element-wise BN to the output of other Rep-FC to ensure training stability.

### B.6 TRAINING DETAILS

Before training, RAT SNN is initialized using fluctuation-driven initialization (Rossbroich et al., 2022), and optimized with the SMORMS3 optimizer combined with cosine learning rate scheduling to accelerate convergence (Rossbroich et al., 2022; Liu et al., 2024). We employed surrogate gradient descent along with a homeostatic activity regularization term to prevent neuron silencing for training (Rossbroich et al., 2022). Surrogate gradient descent minimized the root mean square error (RMSE) between the predicted $\hat{y}_i$ and target finger velocities $y_i$:

$$\text{RMSE} = \sqrt{\frac{1}{N_T} \sum_{i=1}^{N_T} (y_i - \hat{y}_i)^2} \tag{12}$$

where $N_T$ is the total time steps. To prevent neuron silencing during training, we add a homeostatic regularization term (activity lower bound, ALB) to the loss function:

$$\mathcal{L}_{\text{reg}}^{\text{ALB}} = \lambda_{\text{ALB}} \cdot \frac{1}{N} \sum_{i=1}^{N} \left[ \max \left( 0, \phi_{\text{ALB}} - \text{fr}_i \right) \right]^2 \tag{13}$$

where $N$ is the total number of spiking neurons, $\text{fr}_i$ is the firing rate of neuron $i$, $\phi_{\text{ALB}}$ is the lower bound of the firing rate, and $\lambda_{\text{ALB}}$ is the corresponding regularization strength coefficient. The overall loss function is defined as:

$$\mathcal{L} = \mathcal{L}_{\text{RMSE}} + \mathcal{L}_{\text{reg}}^{\text{ALB}} \tag{14}$$

For lightweight finetuning stage, we further impose an AUB constraint on the average firing rate of spiking neurons:

$$\mathcal{L}_{\text{reg}}^{\text{AUB}} = \lambda_{\text{AUB}} \cdot \frac{1}{N} \sum_{i=1}^{N} \left[ \max \left( 0, \text{fr}_i - \phi_{\text{AUB}} \right) \right]^2 \tag{15}$$

$$\mathcal{L}_{\text{light}} = \mathcal{L}_{\text{RMSE}} + \mathcal{L}_{\text{reg}}^{\text{ALB}} + \mathcal{L}_{\text{reg}}^{\text{AUB}} \tag{16}$$

where $\phi_{\text{AUB}}$ is the upper bound of the firing rate and $\lambda_{\text{AUB}}$ is the corresponding regularization strength coefficient.

The number of epochs for cross-condition pretraining is set to 120, while all other training stages adopt early stopping to ensure full convergence. Notably, we equalized input sizes across subjects by replicating shorter recordings. We trained single-session model with 5 random seeds (1-5) to ensure result stability. For pretrained model, we used 2 different random seeds for pretraining, then applied 2 random seeds respectively during the single-session finetuning process, resulting in 4 seed models. All training completed on a single Nvidia 4090D GPU, with cross-condition pretraining taking under two days, and the remaining stages finishing within one day.

### B.7 HYPER-PARAMETERS SETTINGS

For self-attention, we set the number of attention heads to 4, with qkv dimensions $(n, d) = (48, 1)$. We configure RepConv with a three-branch structure using kernel sizes of 3, 1, and 0 (identity mapping), employing padding to ensure the output dimensions equal the input dimensions. The specific hyperparameter settings are shown in Table 8. And following Liu et al. (2024), the parameters for LIF and LI neurons are shown in Table9.

### B.8 BASELINE SET UP

In this work, we compare our method with various baseline models to comprehensively evaluate the decoding performance and energy efficiency of RAT SNN. For the performance of these baseline models on the corresponding datasets, we first use the results reported in the original publications as the baseline performance. We select bigRSNN (Liu et al., 2024) and its pretrained version as the major SNN-based baseline models, and AEGRU (Wang et al., 2024) and POYO (Azabou et al., 2023) as the major ANN-based baseline models, representing the current state-of-the-art in the SNN and ANN fields, respectively. For the results of major baseline models that are not reported in the

Table 8: Hyperparameter Settings for RAT SNN

| Parameter | Value |
|---|---|
| Input size | 192 |
| Attention head num ($n_{\text{head}}$) | 4 |
| Dimension of qkv ($n, d$) | (192, 1) |
| Rep-conv kernel size | 3 |
| Stride | 1 |
| Dilation | 1 |
| Bias | False |
| MLP size | (512,1) |
| Readout head num | 5 |
| learning rate | 1.00E-02 |

Table 9: Time Constants and Training Parameters of LIF and LI Neurons

| Parameter | Value |
|---|---|
| Synaptic time constant (s) | 1.00E-02 |
| LIF membrane time constant (s) | 2.00E-02 |
| LI membrane time constant (s) | 5.00E-02 |
| Time constants heterogeneous | True |
| Time constants trainable | True |
| L2 strength of ALB $\lambda_{\text{ALB}}$ | 100 |
| L2 threshold of ALB $\phi_{\text{ALB}}$ | 1.00E-03 |
| L2 strength of AUB $\lambda_{\text{AUB}}$ | 0.01 |
| L2 threshold of AUB $\phi_{\text{AUB}}$ | 10 |
| Dropout rate | 0.3 |

original papers for certain datasets, we evaluate these baseline models using the testing framework provided by NeuroBench (Yik et al., 2025).

For bigRSNN and AEGRU, we train and test them on the corresponding datasets using their recommended hyperparameters. For POYO, since its testing framework differs from that of NeuroBench, we modified its pretraining procedure according to the NeuroBench testing standard on the NHP dataset, and trained and tested it with the provided hyperparameters.

Since different baseline models adopt different training strategies and preprocessing pipelines (e.g., the random offset sampler in POYO and Gaussian noise augmentation and jitter in AEGRU), it is difficult to define a fully unified preprocessing procedure that aligns all baselines. Therefore, for the baseline models in this paper, we use the hyperparameters and preprocessing strategies recommended by the original authors to ensure that our evaluation faithfully reflects the performance of each method.

### B.9 COMPUTATIONAL COST ESTIMATION

To evaluate the computational cost of RAT SNN and baseline models, we used the computational cost estimation method provided by NeuroBench (Yik et al., 2025). Specifically, we calculated the total effective MACs and ACs on the same test data as metrics for computational cost. Furthermore, following Yin et al. (2021), we set the energy consumption of a single MAC operation to be 31 times that of a single AC operation to estimate the energy efficiency of each model.

The estimation method provided by NeuroBench only considers the computational cost of basic linear and RNN modules in the model, but does not provide a calculation method for attention modules. Therefore, for multi-head attention modules, we first determine whether the data is binary before each matrix multiplication to distinguish between AC and MAC operations. We then binarize the input data, and based on the output of the matrix multiplication, count the number of AC and MAC operations according to the binary determination results.

### B.10 AC-ONLY OPERATIONS IN SELF-ATTENTION

In RAT SNN, the self-attention operates entirely with AC operations. This is achieved by binarizing the Q, K, and V matrices to Qs, Ks, and Vs by LIF layers before performing matrix multiplications. Specifically, according to Eq. (6), we first compute ($K_{s_i}^T V_{s_i}$). Because $K_{s_i}$ and $V_{s_i}$ contain only 0/1 values, the matrix product can be implemented using AC-only operations (the identity element under multiplication leaves any number unchanged, and this property extends to matrix multiplication). When multiplying $Q_{s_i}$ with ($K_{s_i}^T V_{s_i}$), $Q_{s_i}$ is also binary, so this step can likewise be realized by ACs. Thus the entire attention computation avoids MAC operations and uses only ACs, consistent with the spike-driven transformer (Yao et al., 2023a).

Table 10: Performance comparison on NHP dataset, averaged across session 01–03 from Monkey I and L. Values without standard deviations are cited from published results. Results are reported as $R^2 \times 100$. Best performing model is in bold and second best model is underlined.

| Method | Monkey I | | | | Monkey L | | | |
|---|---|---|---|---|---|---|---|---|
| | Session 01 | Session 02 | Session 03 | **I Avg** | Session 01 | Session 02 | Session 03 | **L Avg** |
| KF | 35.00 | 0.60 | 27.80 | 21.13 | 9.20 | 19.60 | 13.80 | 14.20 |
| ANN2d | 63.27 | 52.41 | 62.17 | 59.28 | 57.27 | 47.46 | 62.72 | 55.82 |
| ANN3d | 67.50 | 59.40 | 65.90 | 64.27 | 59.70 | 52.80 | 64.20 | 58.90 |
| GRU | 74.00 | 64.00 | 75.00 | 71.00 | 61.00 | 60.00 | 66.00 | 62.33 |
| AEGRU | 76.00 | **65.00** | 76.00 | 72.00 | 68.00 | 62.00 | 70.00 | 67.00 |
| POYO-SS | 73.02±0.70 | 55.60±1.95 | 58.38±3.44 | 62.33±0.85 | 64.27±2.66 | 57.63±3.50 | 59.70±1.78 | 60.53±2.32 |
| POYO-CS | 77.53±0.73 | 63.70±0.82 | 71.73±0.50 | 70.99±0.42 | 72.42±1.12 | 65.98±1.49 | 70.48±0.77 | 69.63±2.32 |
| SNN2d | 63.30 | 52.40 | 62.20 | 59.30 | 57.30 | 47.50 | 62.70 | 55.83 |
| SNN3d | 69.70 | 57.70 | 65.20 | 64.20 | 62.30 | 56.80 | 68.10 | 62.40 |
| bigRSNN-SS | 77.43±0.37 | 59.17±0.76 | 74.78±0.73 | 70.46±0.24 | 67.24±0.31 | 59.32±1.78 | 69.82±0.81 | 65.40±0.88 |
| bigRSNN-CS | 76.96±0.25 | 58.53±1.42 | 77.17±0.69 | 70.89±0.45 | 70.02±0.66 | 62.69±0.83 | 73.38±0.64 | 68.70±0.36 |
| RAT SNN-SS | 78.06±0.77 | 63.29±0.99 | 75.33±1.01 | 72.22±0.38 | 66.88±2.43 | 60.87±2.51 | 71.15±0.87 | 66.30±0.95 |
| RAT SNN-CS | 79.68±0.29 | 64.07±0.61 | **79.03±0.54** | **74.26±0.14** | 69.12±1.20 | 62.11±0.80 | 74.68±0.56 | 68.63±0.44 |
| RAT SNN-CC | **80.13±0.32** | 63.04±1.04 | 79.01±0.46 | 74.06±0.42 | 68.99±1.80 | **66.16±0.32** | **76.05±0.80** | 70.40±0.68 |
| RAT SNN-CC-P | 78.36±0.57 | 62.61±0.63 | 78.57±0.38 | 73.18±0.17 | **70.05±0.42** | 66.55±1.91 | 75.64±1.56 | **70.75±1.24** |

## C  EVALUATION DETAILS

Due to space limitations in the main text, we provide detailed session-specific decoding results here. For NHP dataset, we selected three representative sessions from Monkey I and Monkey L to evaluate multiple versions of RAT SNN, labeled as Session 01, Session 02, Session 03 (Table10). For private dataset, we selected three random sessions recorded in 2022 from Monkey C05 to evaluate multiple versions of RAT SNN, labeled as C05_1, C05_2, C05_3 (Table11). The specific results for NLB_MAZE and NLB_RTT have already been presented in the main text.

The results demonstrate that all versions of RAT SNN perform excellently on NHP dataset, and as the scope of pretraining data increases, RAT SNN's decoding accuracy gradually improves. This proves that RAT SNN can effectively overcome neural temporal drift and neural variability across different subjects and tasks through cross-session and cross-condition training, thereby enhancing decoding accuracy.

For private dataset, RAT SNN also performs excellently across three sessions of Monkey C05, and when generalizing to the unseen sessions C05_2025, RAT SNN's decoding accuracy remains at the highest level with faster convergence, demonstrating the generalization capability across long temporal scales(Fig.14).

Table 11: Performance comparison ($R^2 \times 100$, mean $\pm$ std) on C05_2022 and C05_2025. "$+$" means generalize result on unseen session.

| Method | Monkey C05_2022 | | | | Monkey C05_2025 | | |
|---|---|---|---|---|---|---|---|
| | Session 01 | Session 02 | Session 03 | **2022 Avg** | Session 01 | Session 02 | **2025 Avg** |
| AEGRU | 77.58±0.39 | 78.15±0.47 | 72.43±0.89 | 76.05±0.43 | 55.07±0.62 | 49.71±1.20 | 52.39±0.81 |
| POYO-SS | 78.22±0.93 | 74.37±0.76 | 78.57±1.52 | 77.05±0.80 | 64.68±1.37 | 65.19±2.91 | 64.93±2.12 |
| bigRSNN-SS | 82.10±0.71 | 85.40±0.28 | 77.64±0.76 | 81.72±0.35 | 65.35±0.89 | 58.96±1.08 | 62.15±0.63 |
| bigRSNN-CS | 84.62±0.84 | 86.19±0.22 | 79.88±0.94 | 83.56±0.54 | 68.41±0.37$^+$ | 64.49±0.29$^+$ | 66.45±2.28$^+$ |
| RAT SNN-SS | 84.24±0.28 | 85.29±0.68 | 75.20±0.88 | 81.58±0.42 | **69.17±0.75** | 64.25±1.01 | 66.71±0.48 |
| RAT SNN-CS | 85.37±0.36 | 87.52±0.38 | 80.13±0.56 | 84.34±0.19 | 68.91±1.03$^+$ | 64.64±1.21$^+$ | 66.77±1.11$^+$ |
| RAT SNN-CC | **85.94±0.21** | **87.73±0.17** | **81.53±0.61** | **85.07±0.21** | 68.21±0.10$^+$ | 64.87±1.59$^+$ | 66.54±0.74$^+$ |
| RAT SNN-CC-P | 84.98±0.32 | 87.29±0.59 | 79.38±0.88 | 83.88±0.11 | 69.09±1.08$^+$ | **65.25±1.29$^+$** | **67.17±1.18$^+$** |

We use Table 12 to compare the decoding accuracy of pretrained RAT SNN models of different scales with RAT SNN-SS on seen sessions across datasets, and Table 13 to compare both decoding accuracy and convergence speed on unseen sessions. Together, these results highlight RAT SNN's ability to improve decoding performance through cross-session and cross-condition pretraining.

We evaluate the generalization capability of RAT SNN-CC to unseen conditions by directly finetuning it on unseen subjects (including both seen tasks and unseen tasks), and compare its performance with the trained-from-scratch version (RAT SNN-SS). The comparison results have been presented

Table 12: Performance comparison on seen sessions.

| Dataset | Session | RAT SNN-SS | RAT SNN-CS | RAT SNN-CC |
|---|---|---|---|---|
| Monkey I | session 1 | 78.06±0.77 | 79.68±0.29 | **80.13±0.32** |
| | session 2 | 63.29±0.99 | **64.07±0.61** | 63.04±1.04 |
| | session 3 | 75.33±1.01 | **79.03±0.54** | 79.01±0.46 |
| Monkey L | session 1 | 66.88±2.43 | **69.12±1.20** | 68.99±1.80 |
| | session 2 | 60.87±2.51 | 62.11±0.80 | **66.16±0.30** |
| | session 3 | 71.15±0.87 | 74.68±0.56 | **76.05±0.80** |
| Monkey C05_2022 | session 1 | 84.24±0.28 | 85.37±0.36 | **85.94±0.21** |
| | session 2 | 85.29±0.86 | 87.52±0.38 | **87.73±0.17** |
| | session 3 | 75.20±0.88 | 80.13±0.56 | **81.53±0.61** |
| Monkey Je (NLB_MAZE) | | 90.76±0.30 | —- | **91.16±0.19** |

Table 13: Performance comparison on unseen sessions.

| Model | Metric | Monkey C05_2025 | | Monkey I |
|---|---|---|---|---|
| | | Session 1 | Session 2 | NLB_RTT |
| RAT SNN-SS | $R^2 \times 100$ | **69.17±0.75** | 64.25±1.01 | 76.34±0.54 |
| | Epoch | 46.40±10.19 | 48.40±6.54 | 171.60±74.82 |
| RAT SNN-CS | $R^2 \times 100$ | 68.91±1.03 | 64.64±1.21 | 78.55±0.19 |
| | Epoch | 28.50±5.07 | 20.25±2.06 | 88.25±39.37 |
| RAT SNN-CC | $R^2 \times 100$ | 68.21±1.00 | **64.87±1.59** | **78.70±0.66** |
| | Epoch | 28.75±4.92 | 21.75±1.50 | 126.75±20.17 |

in the main text. Here we supplement the individual learning curve comparison results for Monkey M, T, and C (Fig.15, Table14).

Table 14: Performance comparison on unseen conditions on sessions used in Fig.5 and Fig.15.

| | Metric | Monkey M | Monkey T | Monkey C |
|---|---|---|---|---|
| RAT SNN-SS | $R^2 \times 100$ | 86.99±0.51 | 74.84±0.32 | 93.21±0.17 |
| | Epoch | 231.56±36.40 | 304.13±64.91 | 364.75±68.26 |
| **RAT SNN-CC** | $R^2 \times 100$ | **87.11±0.15** | **76.29±1.09** | **93.30±0.11** |
| | **Epoch** | **167.38±48.19** | **106.63±16.65** | **221.13±37.28** |

## C.1 COMPARISON WITH SPIKACHU

Spikachu (Mentzelopoulos et al., 2025) is a recently proposed SNN-based neural decoding model that also incorporates attention mechanisms. While its evaluation datasets and strategies differ from ours, on the common NLB_MAZE and NLB_RTT datasets, Spikachu achieved $R^2$ scores of 0.78 and 0.56, respectively, which are lower than our method (RAT SNN-CC-p: 0.9102 and 0.7808).

Notably, Spikachu employs an ANN harmonizer before the SNN module to address neural activity variability across conditions, introducing a substantial number of MAC operations. In contrast, our approach is a pure SNN model that maintains AC-only operations between synapses during inference. Furthermore, RAT SNN features a smaller architecture with fewer layers, resulting in lower inference latency and reduced parameter count. According to the reported results, the cross-condition version of Spikachu (Spikachu-mp) consumes 4.75 $\mu$J and 4.96 $\mu$J on NLB_MAZE and NLB_RTT, respectively. Based on the same energy evaluation metric (Bal & Sengupta, 2024; Zhu et al., 2024), RAT SNN-CC-p consumes significantly less energy, approximately $4351.83 \pm 157.23$ pJ and $8386.47 \pm 1207.11$ pJ on these datasets.

Although Spikachu offers an effective solution for handling cross-condition variability in SNNs and provides valuable insights for model improvement, the trade-offs in energy consumption and

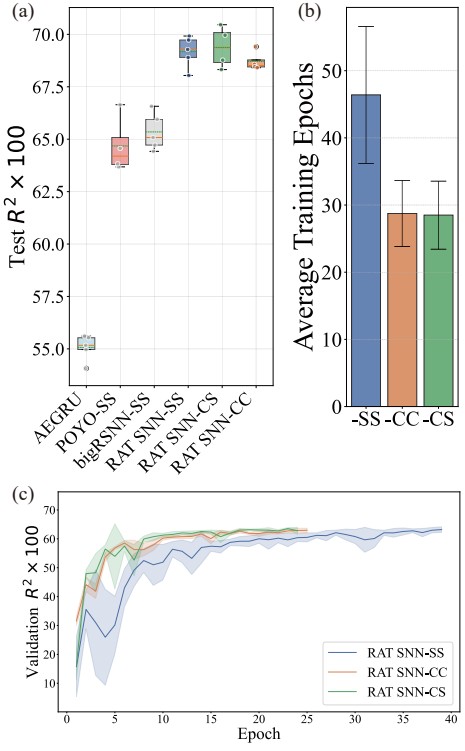

Figure 14: **Result of neural temporal drift generalization on monkey C05_2025.** (a) $R^2$ compared to baseline models. (b) Training epochs required for convergence. (c) Learning curves, with the shaded area denotes std and aligned to the fastest converging seed. Using session 01 as example. [S-S] denotes single-session models, [C-S] cross-session models, and [C-C] cross-condition models.

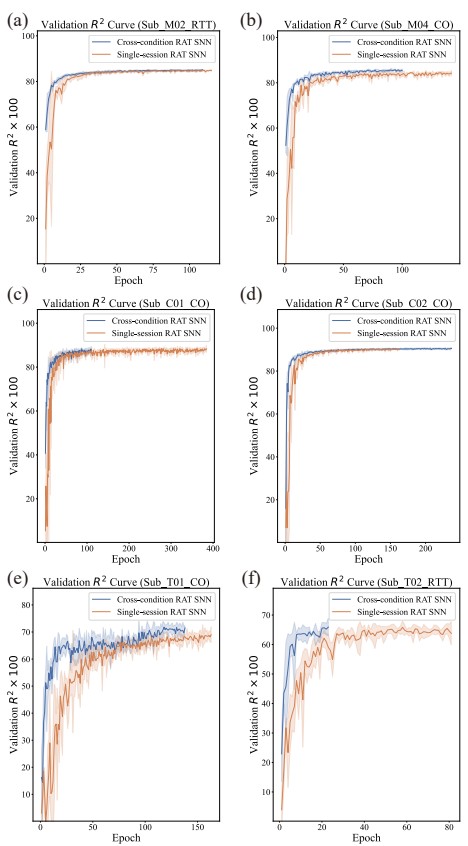

Figure 15: **Supplementary learning curves for Monkey M, T, and C.** The shaded area represents std, and the curve length is aligned to the fastest converging seed. RTT here means random target task with joystick.

decoding accuracy may stem from its deep architecture. A similar phenomenon regarding model depth was also observed in our ablation studies (Section F).

## D    CROSS-CONDITION PRETRAINING EFFECTIVENESS

To rule out the possibility that the effect of cross-condition pretraining is solely due to the increased amount of pretraining data, we designed a controlled experiment comparing RAT SNN-CC with RAT SNN-CS trained on the same amount of pretraining data.

Specifically, we constructed a small-scale cross-condition dataset containing six sessions: "indy_20160915_01", "indy_20160921_01", "indy_20160927_04", "loco_20170301_05", "c05_20220117094031", and "c05_20220120094153", which are from three different subjects (Monkey I, L, and C05), totaling 870,822 timestamps (3,483.29 seconds).

We also constructed a small-scale cross-session dataset containing six sessions: "indy_20160915_01", "indy_20160921_01", "indy_20160927_04", "indy_20160411_02", "indy_20160411_01", and "indy_20160419_01", all from Monkey I, totaling 871,949 timestamps (3,487.80 seconds).

We trained RAT SNN-CC-mini and RAT SNN-CS-mini on these two datasets, respectively, and evaluated them on the unseen Session 01-03 of Monkey I (as recommended by NeuroBench). The

results show that RAT SNN-CC-mini outperforms RAT SNN-CS-mini on the unseen sessions of Monkey I (Table15), demonstrating that the effectiveness of cross-condition pretraining is not solely due to the increased amount of pretraining data. Compared to training on a single condition, introducing a greater diversity of neural activity patterns can more effectively improve the decoding accuracy of RAT SNN.

Table 15: Performance comparison ($R^2 \times 100$, mean $\pm$ std) of RAT SNN-CS-mini and RAT SNN-CC-mini on Monkey I.

| Method | Session 01 | Session 02 | Session 03 | Average |
|---|---|---|---|---|
| RAT SNN-CS-mini | **78.17±0.70** | 62.82±1.23 | 76.08±0.77 | 72.36±0.30 |
| RAT SNN-CC-mini | 78.11±0.22 | **67.92±6.98** | **77.52±0.43** | **74.52±2.45** |

## E    SUPPLEMENT FOR ABLATION STUDY

The above comparative experiments have demonstrated the importance of cross-session pretraining and cross-condition pretraining for improving the decoding accuracy of RAT SNN. To further validate the impact of our design on decoding accuracy, we investigated the effects of each component of RAT SNN on decoding accuracy across both single-session and cross-condition dimensions.

### E.1    ABLATION STUDY ON SINGLE-SESSION RAT SNN

Using the small-scale dataset (Monkey C05) as an example, we conduct ablation experiments on the RAT SNN-SS to evaluate how its design affects decoding accuracy. The ablation study assesses from three perspectives: (1) the overall model architecture, (2) the configuration of LIF layers, and (3) the re-parameterization method.

#### E.1.1    ABLATION STUDY ON THE OVERALL MODEL ARCHITECTURE

We first compare the decoding performance of CST attention block with classical spike-driven transformer block (represented by (Yao et al., 2023a)) on the CST task. Specifically, we design a RAT SNN variant based on the spike-driven transformer block, denoted as SDT SNN, while keeping the remaining architecture identical to RAT SNN. We then evaluate SDT SNN, RAT SNN-192 (where the output dimension of the CST attention block is set to 192, matching that of SDT SNN), and the final version RAT SNN-512 (shown as RAT SNN-SS above), in order to assess how the two attention modules affect decoding accuracy. The ablation results confirm the advantage of RAT SNN's CST attention block in terms of decoding accuracy (80.08 for SDT SNN vs. 80.49 for RAT SNN-192), as well as the performance gains obtained by increasing the output dimension of the attention module (80.49 for RAT SNN-192 vs. 81.58 for RAT SNN-512, Table16).

We additionally construct an SNN model that contains only RepFC-LIF layers, denoted as RepFC SNN, to evaluate the contribution of the attention module to decoding accuracy. Specifically, RepFC SNN-H1 replaces the CST attention block in RAT SNN-512 with a single RepFC-LIF layers of size 512, while RepFC SNN-H3 replaces it with three RepFC-LIF layers of sizes [192, 192, 512], matching the depth and width of RAT SNN. The ablation results show that introducing the attention module substantially improves the decoding accuracy of RAT SNN, indicating that the attention mechanism plays an important role in capturing spatial dependencies in neural activity (81.09 for RepFC SNN-H1 vs. 79.47 for RepFC SNN-H3 vs. 81.58 for RAT SNN-512, Table16).

We further investigate how increasing the depth of the CST attention block affects decoding performance. The results show that a deeper attention block actually degrades the decoding accuracy of RAT SNN (79.53 for RAT SNN-192-A2 vs. 81.58 for RAT SNN-512). Combined with the comparison between RepFC SNN-H3 and RepFC SNN-H1 (81.09 for RepFC SNN-H1 vs. 79.47 for RepFC SNN-H3), this is consistent with our previous findings and indicates that, for the CST task, increasing model depth is detrimental rather than beneficial to improving decoding accuracy (Table16).

In addition, we compare a w/o attention-RepFC variant of RAT SNN, in which the RepFC branch is removed from the CST attention block, with the standard RAT SNN to evaluate the impact of

Table 16: Ablation study on RAT SNN-SS on Monkey C05 ($R^2 \times 100$, mean $\pm$ std).

| | Model | C05_2022 | C05_2025 |
|---|---|---|---|
| Architecture | SDT SNN | 80.09±0.23 | 65.21±0.57 |
| | RAT SNN-192 | 80.49±0.38 | 66.61±0.79 |
| | RAT SNN-192-A2 | 79.53±0.22 | 66.43±0.43 |
| | RepFC SNN-H1 | 81.09±0.40 | 66.26±0.35 |
| | RepFC SNN-H3 | 79.47±0.96 | 65.74±1.00 |
| LIF | w/o syn dynamic | 79.13±0.50 | 58.04±0.52 |
| | w/o recurrent | 79.50±0.45 | 60.48±1.67 |
| | **RAT SNN-512** | **81.58±0.42** | **66.71±0.48** |

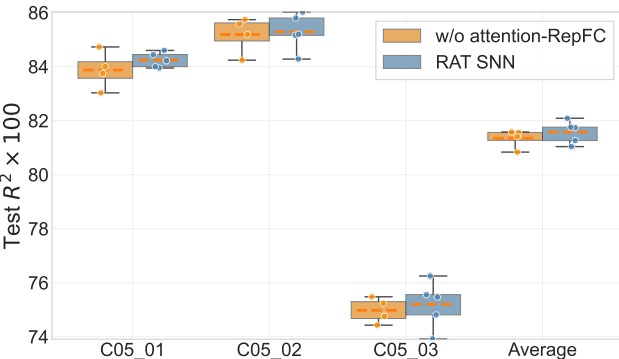

Figure 16: **Performance comparison between the standard RAT SNN and the w/o attention-RepFC version.** The dashed line in the boxplot represents average value..

the RepFC branch on decoding accuracy. The ablation results for attention-RepFC demonstrate that RepFC branch can effectively improve decoding accuracy of RAT SNN. Although the improvement is modest (81.35 vs. 81.58), across all three sessions of Monkey C05, the addition of the RepFC branch leads to varying degrees of improvement in decoding accuracy of RAT SNN, indicating that the performance gains from this strategy are consistent (Fig. 16), Table17).

### E.1.2 Ablation study on the configuration of LIF layers

In this part, we mainly examine the impact of recurrent connections and synaptic dynamics on decoding accuracy (Table16). The ablation results show that both structures provide substantial improvements in decoding performance, with the removal of synaptic dynamics leading to a decrease in decoding accuracy from 81.58 to 79.13, and the removal of recurrent connections resulting in a drop to 79.50.

### E.1.3 Ablation study on the re-parameterization method

This section compares the impact of the re-parameterization method on RAT SNN-SS. Before training, we re-parameterize RAT SNN to obtain a w/o re-parameterization version of RAT SNN that does not include multi-branch Conv and BN structures.

The ablation results for the re-parameterization method demonstrate that this strategy not only effectively improves the learning speed of RAT SNN but also enhances decoding accuracy. Specifically, compared to the standard RAT SNN, the w/o re-parameterization version of RAT SNN shows a 17.16% decrease in decoding accuracy ($R^2$ drops from 81.58±0.42 to 67.58±2.50), while the training epochs required for full convergence increase by approximately 3 times (average training epochs increase from 179±67 to 545±84, Table17).

Table 17: Ablation study results on single-session RAT SNN components. The top performing models in each category are indicated in boldface.

| Method | Avg./$R^2 \times 100$ | Avg./Epoch |
|---|---|---|
| w/o re-param | 67.58±2.50 | 545±84 |
| w/o attention-RepFC | 81.35±0.35 | 213±65 |
| **RAT SNN-SS** | **81.58±0.42** | **179±67** |

Table 18: Performance comparison ($R^2 \times 100$) between subject-specific BN and subject-shared BN on cross-condition RAT SNN. the top performing models in each category are indicated in boldface.

| Session | Shared BN | Specific BN |
|---|---|---|
| C05_1 | 84.27±0.82 | **84.79±0.64** |
| C05_2 | 85.66±1.27 | **86.84±0.52** |
| C05_3 | 77.01±1.85 | **79.05±1.53** |
| C05_Avg. | 82.31±0.98 | **83.56±0.41** |
| I_1 | 78.29±0.57 | **78.30±0.71** |
| I_2 | 60.52±1.81 | **62.16±0.47** |
| I_3 | 75.15±1.21 | **76.35±0.97** |
| I_Avg. | 71.32±0.93 | **72.27±0.66** |
| I_NLB_RTT. | 77.32±0.70 | **77.89±0.63** |
| L_1 | 67.00±1.99 | **68.17±2.00** |
| L_2 | **62.86±1.22** | 57.45±3.94 |
| L_3 | 71.84±0.17 | **72.10±0.77** |
| L_Avg. | **67.23±0.85** | 65.91±1.18 |

## E.2 ABLATION STUDY ON CROSS-CONDITION RAT SNN

For RAT SNN-CC, we primarily compared the decoding accuracy between subject-specific BN and subject-shared BN.

Due to the high computational resource and time requirements for cross-condition training, we used a subset of the cross-condition dataset for comparative experiments. Specifically, for the data from Monkey I, L, and C05 in the cross-condition dataset, we selected only the first 6 sessions from each for comparison, reducing the total session count from 58 to 19. The subsequent finetuning approaches for both subject-specific BN RAT SNN and subject-shared BN RAT SNN remained consistent with the aforementioned methods.

Across the 4 sessions of Monkey C05, subject-specific BN RAT SNN consistently outperforms the subject-shared BN version. In other dataset, subject-specific BN RAT SNN similarly outperforms the subject-shared BN version, with only session 2 from Monkey L showing inferior performance (Table18). This indicating that during the cross-condition pretraining stage, subject-specific BN can better extract the underlying distributions of neural activity under different conditions, thereby improving the decoding accuracy of RAT SNN.

## E.3 ABLATION STUDY ON LIGHTWEIGHT FINETUNING STRATEGY

In this work, we advocate combining direct finetuning with lightweight finetuning and AUB regularization during the lightweight finetuning stage on target session after pretraining. To validate the effectiveness of this finetuning strategy, we compare the results of direct finetuning with lightweight finetuning.

Specifically, we pretrained RAT SNN on the cross-session dataset, then directly finetuned it on the target session to obtain the Pre-LF version of RAT SNN. Subsequently, we performed lightweight finetuning on Pre-LF version to obtain the Post-LF version. We compared these two versions to evaluate the impact of lightweight finetuning on decoding accuracy (Table19). Similarly, we conducted

Table 19: Performance comparison ($R^2 \times 100$) before and after lightweight finetuning across cross-session RAT SNN. Bold indicates the best result in each row.

| Session | Pre-LF | Post-LF |
|---|---|---|
| C05_1 | 84.93±0.60 | **85.37±0.36** |
| C05_2 | 86.41±1.37 | **87.52±0.38** |
| C05_3 | **80.34±2.01** | 80.13±0.56 |
| C05_Avg. | 83.89±0.71 | **84.34±0.19** |
| I_1 | 78.81±0.56 | **79.68±0.29** |
| I_2 | 62.22±1.10 | **64.07±0.61** |
| I_3 | 78.54±0.31 | **79.03±0.54** |
| I_Avg. | 73.19±0.49 | **74.26±0.14** |
| I_NLB_RTT | 78.41±0.77 | **78.55±0.19** |
| L_1 | 66.59±1.50 | **69.12±1.20** |
| L_2 | 59.51±2.48 | **62.11±0.80** |
| L_3 | 74.58±0.68 | **74.68±0.56** |
| L_Avg. | 66.89±0.42 | **68.63±0.44** |

Table 20: Performance comparison ($R^2 \times 100$) before and after lightweight finetuning and after pruning across cross-condition RAT SNN

| Session | Pre-LF | Post-LF | After prune |
|---|---|---|---|
| C05_1 | **86.05±0.15** | 85.94±0.21 | 84.98±0.32 |
| C05_2 | 86.91±0.43 | **87.73±0.17** | 87.29±0.59 |
| C05_3 | **82.19±1.23** | 81.53±0.61 | 79.38±0.88 |
| C05_Avg. | 85.05±0.31 | **85.07±0.21** | 83.88±0.11 |
| I_1 | 79.21±0.39 | **80.13±0.32** | 78.36±0.57 |
| I_2 | 61.96±1.38 | **63.04±1.04** | 62.61±0.63 |
| I_3 | 77.70±0.30 | **79.01±0.46** | 78.57±0.38 |
| I_Avg. | 72.96±0.41 | **74.06±0.42** | 73.18±0.17 |
| I_NLB_RTT | 76.73±0.91 | **78.70±0.66** | 78.08±0.75 |
| L_1 | 67.07±2.59 | 68.99±1.80 | **70.05±0.42** |
| L_2 | 64.65±1.59 | 66.16±0.32 | **66.55±1.91** |
| L_3 | **76.33±1.48** | 76.05±0.80 | 75.64±1.56 |
| L_Avg. | 69.35±1.01 | 70.40±0.68 | **70.75±1.24** |
| NLB_MAZE | 90.95±0.30 | **91.16±0.19** | 91.02±0.15 |

analogous comparative experiments on the cross-condition scene with additional iterative pruning (Table20).

The results demonstrate that lightweight finetuning can effectively improve decoding accuracy across both cross-session and cross-condition scenarios. This demonstrates that RAT SNN can perform more refined search in the local parameter space through lightweight finetuning, thereby improving decoding accuracy. And iterative pruning is no harm to decoding accuracy with effective effect to reduce the computational consumption of RAT SNN.

## F DISCUSSION ON THE DEPTH OF SNN IN CST DECODING TASK

In image-related applications, deep SNN models often achieve better performance than shallow ones. However, in many neuromorphic domains, shallow SNNs actually outperform deeper models, as observed in tasks such as Spiking Heidelberg Digits and DVS-Gesture (Rossbroich et al., 2022). Therefore, in the CST decoding task with pretraining (to minimize the adverse effects of limited dataset size on deep models), we compared SNNs with 1–3 hidden layers and found that, when using the simplest fully connected RSNN architecture, increasing the number of hidden layers not only fails to improve decoding performance but in fact leads to a degradation (Fig. 17).

The potential reasons for this phenomenon may include: (1) Inaccurate gradient estimation: surrogate gradients are used to approximate the non-differentiable spike function, and the approximation

error accumulates more severely as the network depth increases. (2) Downstream neuron death in deep SNN models and increased sensitivity of deeper SNNs to initialization (Rossbroich et al., 2022). (3) Differences in data structure: in natural image tasks, the spatial structure is highly complex, requiring multiple layers of convolutions and nonlinearities to hierarchically extract features from local edges to semantic concepts. In contrast, most neuromorphic benchmarks (including the CST task) typically involve "event streams + relatively simple spatial structure," where the key to decoding often lies in temporal patterns and spiking rhythms rather than rich high-level spatial semantics.

Therefore, in the CST decoding task, using a shallow SNN is in fact a better choice. We compress the depth of the spike transformer block and adopt a single-layer attention structure, making the model more advantageous in terms of both model size and computational operations.

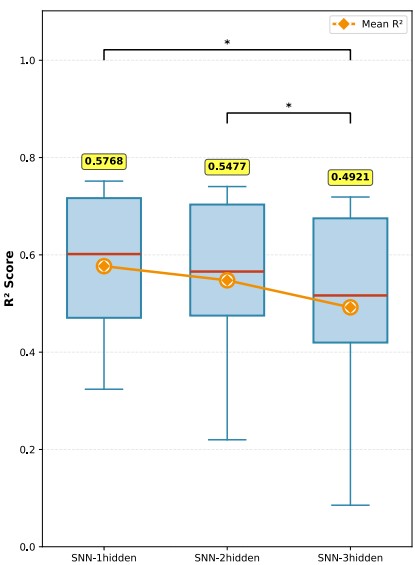

Figure 17: **Performance comparison between 1–3 hidden layers SNN.** These SNN model are tested on Monkey I and Monkey L with random seed 1-5, the hidden LIF layer size is 64. The large variance is caused by differences in CST signals across subjects.

# G   THE USE OF LARGE LANGUAGE MODELS (LLMS)

In this work, all research, ideation, and writing were conducted by the authors. Copilot was used solely as an assistive tool for polishing the manuscript and code annotation. All content generated with Copilot's assistance was thoroughly reviewed and revised by the authors to ensure the originality and quality of the work.

