# OpenReview forum: "Pretraining with Re-parametrized Self-Attention: Unlocking Generalizationin  SNN-Based Neural Decoding Across Time, Brains, and Tasks"
_ICLR.cc/2026/Conference — ICLR 2026 Poster_

### Official Review · Reviewer_c3gQ · 2025-10-23

**Soundness:** 3
**Presentation:** 2
**Contribution:** 2
**Rating:** 6
**Confidence:** 2

**Summary:**

This paper proposes RAT SNN, a Spiking Neural Network integrating re-parameterized self-attention with a cross-condition pre-training framework. It aims to provide a neural decoding solution for iBMI that offers high accuracy, strong generalization capabilities, and high energy efficiency.

**Strengths:**

The paper's strength lies in its ingenious integration of various advanced technologies to address a critical and highly practical problem (efficiency and generalization bottlenecks in iBMI). It further demonstrates an excellent balance between decoding accuracy and computational efficiency through extremely detailed and convincing experiments.

**Weaknesses:**

I am not familiar with the iBMI field, so I can only raise the following weaknesses from the SNN perspective.

1.The method primarily represents a sophisticated combination and engineering optimization of existing techniques (e.g., RepVGG-style re-parameterization, spiking Transformers, and LIF layer with recurrent connections) within a specific application scenario, rather than proposing an entirely new algorithm or theoretical framework.Weakness

2.During the inference process of the attention module, how does matrix multiplication avoid MAC operations? Please provide a more detailed explanation.

3.Is the use of the attention mechanism effective on a network with only 4 layers? The paper lacks corresponding ablation experiments, such as comparing with a regular SNN that uses reparameterization.

**Questions:**

It is strongly recommended that the authors move the reparameterization effectiveness experiments from the appendix to the main body of the paper.

---

> ### Author Response · Authors · 2025-11-20
> **Response to Reviewer c3gQ on Weakness 1 (1/2)**
>
> We appreciate your thorough review of our work and the thoughtful, constructive feedback.
>
> ## **Weakness 1:**
> >The method primarily represents a sophisticated combination and engineering optimization of existing techniques (e.g., RepVGG-style re-parameterization, spiking Transformers, and LIF layer with recurrent connections) within a specific application scenario, rather than proposing an entirely new algorithm or theoretical framework.
>
> We would like to clarify the scope and positioning of our contribution and innovation on the architectural side.
> In particular, the proposed RAT SNN is, to the best of our knowledge, the first spike‑based self‑attention architecture specifically designed for cortical spike train (CST) decoding in iBMI settings.
> We fully agree that both re‑parameterization and self‑attention have been explored in prior SNN transformer works [1–4]. However, directly adopting multi‑layer transformer blocks from these works leads to large model size and high computational cost, and in our CST decoding experiments it actually degrades decoding accuracy.
> **With the strict energy constraints of iBMI systems，we redesign a task‑specific, lightweight architecture and propose a CST re‑parametrized attention block**.
> Together with the stepwise training framework, this design enables our model to outperform strong SNN baselines and to match state‑of‑the‑art ANN models in decoding accuracy (Tables 2–4), while achieving much lower computational cost (Table 7) and significantly fewer parameters (Fig. 7).
>
> To demonstrate the effectiveness of the proposed architecture and the computational advantages brought by re‑parameterization at inference phase, we provide a detailed description of the re‑parameterization process in the paper.
>
> In light of your suggestions, and to improve the readability of the paper and better highlight our main contributions, in the revised version we have added a more detailed description of the CST re‑parametrized attention block and reorganized the structure of Section 3.2 in the Method part.
>
> In summary, we emphasize: (i) the lightweight redesign of the prior spike-driven transformer , (ii) the parallel Conv+FC Q/K/V branches for capturing both local and global features under discontinuous electrode channel indices, and (iii) the introduction of a linear mapping on the spike-element-wise shortcut, which enables rescaling of the output layer of attention block.
>
> >**Detail description of the CST re‑parametrized attention structure.**
> >
> >Previous experiments have shown that overly deep SNNs can impair decoding accuracy and real-time efficiency in CST decoding for BCI systems and other applications [6].
> >Therefore, we merge the MLP block and the output layer in the attention block from prior spike-driven transformer designs [1–4], reducing the depth from the classic five layers to three; this shallower structure not only helps improve accuracy but also reduces model size and computational operations.
> >
> >To further reduce model depth, we compute Qs/Ks/Vs directly from the input CST spikes and, since they lack membrane potentials, replace the membrane-shortcut (MS) in spike-driven transformers with a spike-element-wise shortcut [5] followed by a linear layer (a "surrogate shortcut"):
> >$$
> >    U_ s = \text{RLIF}(\text{Attention}_ s + \text{Rep-FC}(X_{CST}))
> >$$
> >where $ U_s $ denotes the output spikes of CST re-parametrized attention block at current time step. Surrogate shortcut enables changing the attention output dimension ($\text{MLP Size}>n\times{d}$, Fig.1), providing flexible output dimensionality and improving performance without increasing depth.
> >
> >Previous studies have shown that convolution layer (Conv) can effectively enhance the performance of spike-driven attentions [3]. However, considering the discontinuity of electrode channel indices and inter-subject differences, we integrate parallel Conv and fully connected layers (FC) for Qs/Ks/Vs, where FC captures global features and Conv focuses on local features across CST channels:
> >$$
> >    Q_s/K_s/V_s = \text{RLIF}(\text{Rep-Conv}(X_{CST}) + \text{Rep-FC}(X_{CST}))
> >$$
> >where $ X_{CST} $ denotes the input CST spikes at current time step and $ \text{Rep-Conv}(\cdot) $ and $ \text{Rep-FC}(\cdot) $ denote the re-parameterized Conv and FC.

---

> > ### Author Response · Authors · 2025-11-20
> > **Response to Reviewer c3gQ on Weakness 1 (2/2)**
> >
> > **Discussion on the depth of SNN in CST decoding task.**
> >
> > We have added Appendix E.4, *“Discussion on the depth of SNN in CST decoding task”*, to discuss the adverse effects of SNN depth on CST decoding, analyze the potential underlying causes, and provide additional preliminary experimental results as theoretical support for the architectural design of our model:
> >
> > >In image-related applications, deep SNN models often achieve better performance than shallow ones. However, in many neuromorphic domains, shallow SNNs actually outperform deeper models, as observed in tasks such as Spiking Heidelberg Digits and DVS-Gesture [6]. Therefore, in the CST decoding task with pretraining (to minimize the adverse effects of limited dataset size on deep models), we compared SNNs with 1–3 hidden layers and found that, when using the simplest fully connected RSNN architecture, increasing the number of hidden layers not only fails to improve decoding performance but in fact leads to a degradation.
> > >
> > >**Table. Performance comparison ($R^2\times100$) between 1–3 hidden layers SNN.**
> > >These SNN model are tested on Monkey I and Monkey L with random seed 1-5, the hidden LIF layer size is 64.
> > >| Model       | Monkey I & L Avg. |
> > >|------------|--------------------|
> > >| SNN-1hidden | 57.68±0.93       |
> > >| SNN-2hidden | 54.77±1.44       |
> > >| SNN-3hidden | 49.21±0.84       |
> > >
> > >The potential reasons for this phenomenon may include:
> > >1. Inaccurate gradient estimation: surrogate gradients are used to approximate the non-differentiable spike function, and the approximation error accumulates more severely as the network depth increases.
> > >2. Downstream neuron death in deep SNN models and increased sensitivity of deeper SNNs to initialization.
> > >3. Differences in data structure: in natural image tasks, the spatial structure is highly complex, requiring multiple layers of convolutions and nonlinearities to hierarchically extract features from local edges to semantic concepts. In contrast, most neuromorphic benchmarks (including the CST task) typically involve "event streams + relatively simple spatial structure", where the key to decoding often lies in temporal patterns and spiking rhythms rather than rich high-level spatial semantics.
> > >
> > >Therefore, **in the CST decoding task, using a shallow SNN is in fact a better choice**. We compress the depth of the spike transformer block and adopt a single-layer attention structure, making the model more advantageous in terms of both model size and computational operations.

---

> ### Author Response · Authors · 2025-11-20
> **Response to Reviewer c3gQ on Weakness 2 and 3**
>
> ## **Weakness 2:**
> >During the inference process of the attention module, how does matrix multiplication avoid MAC operations? Please provide a more detailed explanation.
>
> As output of LIF layer, the queries, keys, and values $(Q_s, K_s, V_s)$ are all binary tensors containing only 0/1 spikes:
> $$
>     Q_s/K_s/V_s = \text{RLIF}(Q/K/V)
> $$
> Since the LIF function (denoted as $\text{RLIF}(\cdot)$ in paper) introduces strong nonlinearity through a threshold-triggered mechanism and enables feature selection and competition similar to Softmax via the term $(Q_{s_i} K_{s_i}^\top)$, it can be viewed as a kernel function that replaces Softmax in **linear attention** [4]. Thus, we obtain:
> $$
> \mathrm{Attention}_ {s_i}
> = \mathrm{RLIF}\big((Q_{s_i} K_{s_i}^\top) V_{s_i}\big)
> = \mathrm{RLIF}\big(Q_{s_i} (K_{s_i}^\top V_{s_i})\big).
> $$
>
> We first compute the term $(K_{s_i}^\top V_{s_i})$. Since both $(K_{s_i})$ and $(V_{s_i})$ are binary (0/1), each multiplication in this matrix product is equivalent to either adding the operand (when the bit is 1) or adding zero (when the bit is 0). Therefore, this matrix multiplication can be implemented purely as AC operations without using explicit multipliers.
>
> Next, we multiply $(Q_{s_i})$ with the pre-computed matrix $(K_{s_i}^\top V_{s_i})$. Again, because $(Q_{s_i})$ is also binary, this step can also be realized only with additions and no MAC units. As a result, the entire attention computation in the spiking domain can be executed using AC-only operations instead of MAC operations.
>
> For clarity, we have added a detailed explanation of these AC-only implementations to Appendix B.10, titled “AC-only operations in self-attention”.
>
>
> ## **Weakness 3:**
> >Is the use of the attention mechanism effective on a network with only 4 layers? The paper lacks corresponding ablation experiments, such as comparing with a regular SNN that uses reparameterization.
>
> Thank you for the suggestion. In the revised manuscript, we have added ablation experiments that compare the RAT SNN model with:
> 1. a regular SNN with the same hidden dimensionality and a regular SNN with the same network depth (as you suggested),
> 2. a classic spiking Transformer block,
>
> to evaluate the effectiveness of our CST re-parameterized attention block:
> >Using the small-scale dataset (Monkey C05) as an example, we conduct ablation experiments on the RAT SNN-SS to evaluate how CST re-parameterized attention block affects decoding accuracy.
> >
> >Comparing our CST attention block (MLP size=512) with the classical spike transformer block (SDT SNN) and RAT SNN‑192 (MLP size=192, the same output dimension as the classical spike transformer block and equal to the input size), our method yields higher decoding accuracy, and enlarging its output dimension via the “surrogate shortcut” further improves performance.
> >
> >Also, compared with the fully connected recurrent SNN baseline (RepFC SNN), our method outperforms both the single-hidden-layer (FC SNN-H1) and 3-hidden-layer (FC SNN-H3) variants, showing the effectiveness of CST attention block in capturing spatial dependencies in CST .
> >
> >**Table. Ablation study results on RAT SNN-SS on Monkey C05.**
> >| Model        | C05_2022          | C05_2025          |
> >|-------------|-------------------|-------------------|
> >| SDT SNN     | 80.09±0.23        | 66.34±0.38        |
> >| RAT SNN-192 | 80.49±0.38        | 68.76±1.12        |
> >| FC SNN-H1   | 81.09±0.40        | 68.54±0.84        |
> >| FC SNN-H3   | 79.47±0.96        | 67.99±1.35        |
> >| **RAT SNN-SS** | **81.58±0.42** | **69.17±0.36**    |

---

> > ### Author Response · Authors · 2025-11-20
> > **Response to Reviewer c3gQ on Question**
> >
> > ## **Question:**
> > >It is strongly recommended that the authors move the reparameterization effectiveness experiments from the appendix to the main body of the paper.
> >
> > Thank you for the suggestion. We have moved the reparameterization effectiveness experiments from the appendix to the main body of the paper and included the ablation studies mentioned above.
> >
> >
> >
> > ## Summary
> >
> > We appreciate you again for the detailed review. If you still has any remaining questions or concerns, we would be glad to hear that.
> >
> > [1] Zhou Z, Zhu Y, He C, et al. Spikformer: When spiking neural network meets transformer[J]. arXiv preprint arXiv:2209.15425, 2022.
> >
> > [2] Zhang J, Tang L, Yu Z, et al. Spike transformer: Monocular depth estimation for spiking camera[C]//European Conference on Computer Vision. Cham: Springer Nature Switzerland, 2022: 34-52.
> >
> > [3] Yao M, Hu J K, Hu T, et al. Spike-driven transformer v2: Meta spiking neural network architecture inspiring the design of next-generation neuromorphic chips[J]. arXiv preprint arXiv:2404.03663, 2024.
> >
> > [4] Yao M, Hu J, Zhou Z, et al. Spike-driven transformer[J]. Advances in neural information processing systems, 2023, 36: 64043-64058.
> >
> > [5] Fang W, Yu Z, Chen Y, et al. Deep residual learning in spiking neural networks[J]. Advances in Neural Information Processing Systems, 2021, 34: 21056-21069.
> >
> > [6] Rossbroich J, Gygax J, Zenke F. Fluctuation-driven initialization for spiking neural network training[J]. Neuromorphic Computing and Engineering, 2022, 2(4): 044016.

---

> > > ### Comment · Reviewer_c3gQ · 2025-11-25
> > >
> > > Thank you for your detailed response and the revisions of the paper, which effectively addressed my concerns. I have no further questions.

---

> > > > ### Author Response · Authors · 2025-11-27
> > > > **Response to Reviewer c3gQ**
> > > >
> > > > We are delighted that your concerns have been fully addressed. We sincerely appreciate the time and effort you have devoted to reviewing our work, which has greatly improved the quality of our paper.
> > > > And, if possible,  we would be very grateful if you could consider reflecting this in your final evaluation (e.g., rating and/or confidence). Thanks again!
> > > >
> > > > If you have any further questions or suggestions, we would be very happy to discuss them and will respond promptly.

---

### Official Review · Reviewer_tbCB · 2025-10-27

**Soundness:** 2
**Presentation:** 3
**Contribution:** 3
**Rating:** 4
**Confidence:** 3

**Summary:**

This paper proposes a re-parameterized self-attention spiking neural network (SNN) architecture for neural signal decoding.
The network employs spiking neurons to process cortical spike train signals, achieving lightweight structure and low-power consumption.
Furthermore, based on existing datasets, they construct a hybrid dataset that integrates cross-condition, cross-session, and cross-subject data, and conduct pretraining on this dataset to alleviate the generalization problem commonly faced by neural decoding models.

**Strengths:**

1. This paper provides a well-grounded analysis of the motivation, clearly identifying that the major challenge in current brain-signal decoding models lies in their poor generalization ability. It also highlights the low-power and lightweight model requirements in edge-side deployment scenarios. Building upon this, the paper logically introduces the use of spiking neural networks (SNNs), which are not only inherently well-suited for processing cortical spike train (CST) signals, but also naturally meet these computational constraints, forming a coherent and well-structured line of reasoning.

2. The experiments in this paper are thorough. A hybrid dataset was specifically designed to meet the requirements of pretraining, and extensive evaluations were conducted across multiple datasets.

**Weaknesses:**

1. The authors’ innovations in training methodology and dataset integration are indeed appealing; however, the network architecture design lacks sufficient novelty. For instance, the re-parameterization operation that fuses Batch Normalization (BN) with Convolution/FC layers has already been widely adopted in CNN-based SNN architectures. Similarly, the multi-branch convolutional structure inspired by RepVGG and its inference-time merging into a single convolution are also common practices in the ANN domain. In this paper, the authors merely apply these existing techniques, yet devote substantial space to describing the re-parameterization process, even emphasizing the “re-parameterized self-attention” in the title—while no genuinely innovative architectural design is evident in this aspect.

2. The variable notation is inconsistent. In the annotation of Figure 1, it is stated that $n_\text{input}$ represents the input dimension, which suggests that n should denote a dimensional quantity. However, later in the text, the authors write that “$n$ and $d$ represent the size of the LIF layers,” a statement whose meaning is unclear—particularly regarding which symbol refers to the number of layers. Furthermore, in Section 3.2, when analyzing the SNN self-attention mechanism, the paper claims that the computational complexity is reduced from O($n^2d$) to O($nd^2$). In this context, $n$ is evidently used to denote the number of tokens, while $d corresponds to the feature dimension, which contradicts the earlier definition. Overall, the inconsistent and ambiguous use of symbols significantly affects the clarity and readability of the paper.

3. The paper provides insufficient analysis of the neuron design. In the design of LIF neurons, the authors propose R-LIF, which introduces a recurrent structure across time to enhance the extraction of temporal information. Specifically, a temporal skip connection is added to integrate past spike firings and synaptic inputs. However, the paper does not provide a detailed analysis of the motivation behind this design, nor does it discuss any possible biological justification for it. Moreover, such an operation is not novel in the context of SNN neuron design.

**Questions:**

1. The authors state that the proposed model uses only four layers of LIF neurons, yet the paper does not provide a detailed description of the overall network architecture. Moreover, a four-layer configuration is considered very shallow for a typical SNN. The fact that the authors claim to achieve SOTA performance in various aspects—and even outperform ANN-based models—with such a shallow network raises doubts about the fairness of the comparison.

2. The authors present the iterative equations for LIF neurons in Equations (1)–(3), but the formulas appear to be incorrect. In Equation (1), it should be $V_{syn}[t]$.

3. In Section 3.2, the authors state that the re-parameterized channel attention consists of re-parameterized convolutional and FC layers. However, I disagree with this claim and believe it represents over-packaging of the concept. In my view, an attention mechanism should dynamically adjust its focus based on the input, helping the model emphasize important parts while filtering out irrelevant information. Moreover, channel attention is a well-established concept introduced many years ago, typically involving the use of convolution to generate attention scores across channels, which then guide the model to reduce redundancy and enhance salient features. Therefore, I do not consider a module composed merely of convolutional and fully connected layers to qualify as a genuine channel attention mechanism.

4. At the end of Section 3.2, the authors mention that global BN is applied to the output of the multi-head attention mechanism. However, I do not clearly understand what the authors mean by global BN. If their intention is to distinguish global BN from head-wise BN, then I wonder—haven’t previous works that employ self-attention already used what could be considered global BN by default? Moreover, has head-wise BN ever actually been used or reported in prior studies?

5. Throughout the paper, the authors repeatedly emphasize the concept of cross-condition when constructing datasets and conducting generalization tests. However, they do not clearly define what exactly a condition refers to. Is it a concept parallel to cross-subject and cross-session, or does it represent a broader notion that encompasses cross-subject, cross-session, and other variations?

6. What are the model parameters and the time step length used in the SNN in this paper? Additionally, how is the time step defined or segmented for the CST signals?

---

> ### Author Response · Authors · 2025-11-20
> **Response to Reviewer tbCB on Weakness 1 (1/2)**
>
> We appreciate your careful reading of our manuscript and the detailed, constructive feedback.
>
> ## **Weakness 1:**
> We sincerely thank you for acknowledging our innovations in training methodology and dataset integration.
>
> Regarding the concern that **the network architecture lacks sufficient novelty**, we would like to clarify the scope and positioning of our contribution and innovation on the architectural side.
> In particular, the proposed RAT SNN is, to the best of our knowledge, the first spike‑based self‑attention architecture specifically designed for cortical spike train (CST) decoding in iBMI settings.
> We fully agree that both re‑parameterization and self‑attention have been explored in prior SNN transformer works [1–4]. However, directly adopting multi‑layer transformer blocks from these works leads to large model size and high computational cost, and in our CST decoding experiments it actually degrades decoding accuracy.
> **With the strict energy constraints of iBMI systems，we redesign a task‑specific, lightweight architecture and propose a CST re‑parametrized attention block**. (replacing the “re‑parameterized channel attention block” term in the revised version, following your and other reviewers’ suggestions).
> Together with the stepwise training framework, this design enables our model to outperform strong SNN baselines and to match state‑of‑the‑art ANN models in decoding accuracy (Tables 2–4), while achieving much lower computational cost (Table 7) and significantly fewer parameters (Fig. 7).
>
> To demonstrate the effectiveness of the proposed architecture and the computational advantages brought by re‑parameterization at inference phase, we provide a detailed description of the re‑parameterization process in the paper.
>
> In light of your suggestions, and to improve the readability of the paper and better highlight our main contributions, in the revised version we have added a more detailed description of the CST re‑parametrized attention block and reorganized the structure of Section 3.2 in the Method part.
> In summary, we emphasize: (i) the lightweight redesign of the prior spike-driven transformer , (ii) the parallel Conv+FC Q/K/V branches for capturing both local and global features under discontinuous electrode channel indices, and (iii) the introduction of a linear mapping on the spike-element-wise shortcut, which enables rescaling of the output layer of attention block.
>
> >**Detail description of the CST re‑parametrized attention structure.**
> >
> >Previous experiments have shown that overly deep SNNs can impair decoding accuracy and real-time efficiency in CST decoding for BCI systems and other applications [6].
> >Therefore, we merge the MLP block and the output layer in the attention block from prior spike-driven transformer designs [1–4], reducing the depth from the classic five layers to three; this shallower structure not only helps improve accuracy but also reduces model size and computational operations.
> >
> >To further reduce model depth, we compute Qs/Ks/Vs directly from the input CST spikes and, since they lack membrane potentials, replace the membrane-shortcut (MS) in spike-driven transformers with a spike-element-wise shortcut [5] followed by a linear layer (a "surrogate shortcut"):
> >$$
> >    U_ s = \text{RLIF}(\text{Attention}_ s + \text{Rep-FC}(X_{CST}))
> >$$
> >where $ U_s $ denotes the output spikes of CST re-parametrized attention block at current time step. Surrogate shortcut enables changing the attention output dimension ($\text{MLP Size}>n\times{d}$, Fig.1), providing flexible output dimensionality and improving performance without increasing depth.
> >
> >Previous studies have shown that convolution layer (Conv) can effectively enhance the performance of spike-driven attentions [3]. However, considering the discontinuity of electrode channel indices and inter-subject differences, we integrate parallel Conv and fully connected layers (FC) for Qs/Ks/Vs, where FC captures global features and Conv focuses on local features across CST channels:
> >$$
>     Q_s/K_s/V_s = \text{RLIF}(\text{Rep-Conv}(X_{CST}) + \text{Rep-FC}(X_{CST}))
> >$$
> >where $ X_{CST} $ denotes the input CST spikes at current time step and $ \text{Rep-Conv}(\cdot) $ and $ \text{Rep-FC}(\cdot) $ denote the re-parameterized Conv and FC.

---

> > ### Author Response · Authors · 2025-11-20
> > **Response to Reviewer tbCB on Weakness 1 (2/2)**
> >
> > **Structural Ablation Study.**
> > We have added a new Section 4.5, “Structural Ablation Study,” in the revised version to describe and validate the effectiveness of our proposed method:
> >
> > >Using the small-scale dataset (Monkey C05) as an example, we conduct ablation experiments on the RAT SNN-SS to evaluate how CST re-parameterized attention block affects decoding accuracy.
> > >
> > >Comparing our CST attention block (MLP size=512) with the classical spike transformer block (SDT SNN) and RAT SNN‑192 (MLP size=192, the same output dimension as the classical spike transformer block and equal to the input size), our method yields higher decoding accuracy, and enlarging its output dimension via the “surrogate shortcut” further improves performance.
> > >
> > >Also, compared with the fully connected recurrent SNN baseline (RepFC SNN), our method outperforms both the single-hidden-layer (FC SNN-H1) and 3-hidden-layer (FC SNN-H3) variants, showing the effectiveness of CST attention block in capturing spatial dependencies in CST .
> > >
> > >**Table. Ablation study results on RAT SNN-SS on Monkey C05.**
> > >| Model        | C05_2022          | C05_2025          |
> > >|-------------|-------------------|-------------------|
> > >| SDT SNN     | 80.09±0.23        | 66.34±0.38        |
> > >| RAT SNN-192 | 80.49±0.38        | 68.76±1.12        |
> > >| FC SNN-H1   | 81.09±0.40        | 68.54±0.84        |
> > >| FC SNN-H3   | 79.47±0.96        | 67.99±1.35        |
> > >| **RAT SNN-SS** | **81.58±0.42** | **69.17±0.36**    |
> >
> > ****
> >
> > **Discussion on the depth of SNN in CST decoding task.**
> >
> > We have added Appendix E.4, *“Discussion on the depth of SNN in CST decoding task”*, to discuss the adverse effects of SNN depth on CST decoding, analyze the potential underlying causes, and provide additional preliminary experimental results as theoretical support for the architectural design of our model:
> >
> > >In image-related applications, deep SNN models often achieve better performance than shallow ones. However, in many neuromorphic domains, shallow SNNs actually outperform deeper models, as observed in tasks such as Spiking Heidelberg Digits and DVS-Gesture [6]. Therefore, in the CST decoding task with pretraining (to minimize the adverse effects of limited dataset size on deep models), we compared SNNs with 1–3 hidden layers and found that, when using the simplest fully connected RSNN architecture, increasing the number of hidden layers not only fails to improve decoding performance but in fact leads to a degradation.
> > >
> > >**Table. Performance comparison ($R^2\times100$) between 1–3 hidden layers SNN.**
> > >These SNN model are tested on Monkey I and Monkey L with random seed 1-5, the hidden LIF layer size is 64.
> > >| Model       | Monkey I & L Avg. |
> > >|------------|--------------------|
> > >| SNN-1hidden | 57.68±0.93       |
> > >| SNN-2hidden | 54.77±1.44       |
> > >| SNN-3hidden | 49.21±0.84       |
> > >
> > >The potential reasons for this phenomenon may include:
> > >1. Inaccurate gradient estimation: surrogate gradients are used to approximate the non-differentiable spike function, and the approximation error accumulates more severely as the network depth increases.
> > >2. Downstream neuron death in deep SNN models and increased sensitivity of deeper SNNs to initialization.
> > >3. Differences in data structure: in natural image tasks, the spatial structure is highly complex, requiring multiple layers of convolutions and nonlinearities to hierarchically extract features from local edges to semantic concepts. In contrast, most neuromorphic benchmarks (including the CST task) typically involve "event streams + relatively simple spatial structure", where the key to decoding often lies in temporal patterns and spiking rhythms rather than rich high-level spatial semantics.
> > >
> > >Therefore, **in the CST decoding task, using a shallow SNN is in fact a better choice**. We compress the depth of the spike transformer block and adopt a single-layer attention structure, making the model more advantageous in terms of both model size and computational operations.

---

> ### Author Response · Authors · 2025-11-20
> **Response to Reviewer tbCB on Weakness 2 and 3**
>
> ## **Weakness 2:**
> We thank you for pointing out the issue that **the variable notation is inconsistent.** Your understanding of $n$ and $d$ in Section 3.2 is exactly correct:
> - $n$ denotes the number of tokens in the attention mechanism,
> - $d$ denotes the feature dimension.
>
> The confusion mainly comes from the annotation in Fig. 1. There we first used subscripted notations for other variables (e.g., $n_{\text{input}}$) and then used a vague sentence:
> >“$n$ and $d$ represent the size of the LIF layers”,
>
> This indeed makes the notation ambiguous.
> In the revised version, we have **redesigned Fig. 1 and clarified the labels**:
> - **“CST Channel Num”** to denote the input dimensionality,
> - **“MLP Size”** to denote the output dimensionality of the CST re‑parametrized attention block.
>
> Each “LIF layer” in Fig. 1 actually corresponds to a **single LIF layer**, so we do not introduce an extra symbol for the number of layers.
>
> **Detailed explanation of (n) and (d):**
> The phrase “$n$ and $d$ represent the size of the LIF layers” in Fig. 1 is intended to mean that this LIF layer has the **same size as the attention output**, which is also the same as the size of $Q/K/V$. Concretely, $Q$, $K$, and $V$ have shape $n, d$, where $n$ is the number of tokens and $d$ is the feature dimension. After passing through the LIF layer, they become spike tensors $Q_s$, $K_s$, and $V_s$ with the **same shape** $n, d$, which are then used to compute the linear attention as in Eq. (6):
> $$
> \mathrm{Attention}_ {s_i} = \mathrm{RLIF}\big((Q_ {s_i} K_{s_i}^T) V_{s_i}\big) = \mathrm{RLIF}\big(Q_{s_i}(K_{s_i}^T V_{s_i})\big).
> $$
>
> Since the feature dimension $d$ is much smaller than the number of tokens $n$, the complexity is reduced from $O(n^2 d)$ for $(Q_{s_i} K_{s_i}^T) V_{s_i}$ to $O(n d^2)$ for $Q_{s_i}(K_{s_i}^T V_{s_i})$, which matches the complexity analysis in Section 3.2.
>
> **Revised explanation of Figure 1:**
> RAT SNN overall architecture.
> Each time step, RAT SNN takes spikes input with dimension $(\text{CST Channel Num},1)$ from cortical spike train (CST), and outputs kinematic predictions after processing through CST re-parametrized attention block and readout block.
> Rep-FC denotes a re-parameterized fully connected layer, Rep-Conv refers to a three-branch 1D convolutional layer.
> $n$ and $d$ represent the number of tokens and the feature dimension of self-attention, MLP Size indicating output dimension of CST re-parametrized attention block. Head and Ro indicates number of attention heads and multi-readout block.
>
>
>
>
>
>
>
>
>
>
> ## **Weakness 3:**
> Thanks for bringing up this question, regarding **the recurrent structure in the R‑LIF neuron**, we would like to clarify both its **biological motivation** in our model. From a biological plausibility perspective, we treat the LIF layers that are connected by linear layers as different brain areas. In this view：
> - the linear layers **approximate long‑range projection pathways between brain areas**,
> - the **recurrent structure within R‑LIF is intended to mimic local microcircuits inside each area**, and at the same time strengthens the model’s ability to capture temporal dependencies.
>
> Combined with the synaptic‑dynamics mechanism, We believe that, in the context of decoding real neural signals, it may be a good strategy to make the model as biologically realistic as possible.
>
> Recurrent structures have been shown to be effective in many previous studies [6]. Therefore, we combine them with synaptic‑dynamics LIF neurons (which we refer to as R‑LIF), and we would like to emphasize that, to the best of our knowledge, **the effectiveness of integrating R‑LIF with self‑attention has not been reported before**.
>
> To support this design, we added an ablation study in Appendix E.1.2 that separately evaluates the contributions of synaptic dynamics and recurrent connections.
> >The ablation results show that both structures provide substantial improvements in decoding performance, with the removal of synaptic dynamics leading to a decrease in decoding accuracy from 81.58 to 79.13, and the removal of recurrent connections resulting in a drop to 79.50.
> >
> >**Table. Ablation study on RAT SNN-SS on Monkey C05 ($R^2\times100$, mean ± std).**
> >| Model           | C05_2022        | C05_2025        |
> >|-----------------|-----------------|-----------------|
> >| w/o syn dynamic | 79.13±0.50      | 63.70±2.67      |
> >| w/o recurrent   | 79.50±0.45      | 60.84±0.10      |
> >| **RAT SNN-512** | **81.58±0.42**  | **69.17±0.36**  |

---

> > ### Author Response · Authors · 2025-11-20
> > **Response to Reviewer tbCB on Question 1-3**
> >
> > ## **Question 1**
> > A detailed description of the overall network architecture, as well as our discussion on the appropriate depth of SNNs for CST decoding, can be found in our response to Weakness 1. Here, we focus on **the fairness of the experimental comparisons**.
> >
> > We would like to clarify how the baselines are reported and evaluated.
> > * For baselines whose results do not include standard deviations in our tables, we directly adopt the numbers reported in their original papers. Specifically, the GRU and AEGRU results in Table 2 are reported in AEGRU [7], and the NDT‑POYO‑1 results in Table 4 are reported in POYO [8].
> > * For baselines we evaluate ourselves (e.g., POYO and bigRSNN in Table 2), we strictly follow the hyperparameter settings recommended by their official open‑source implementations. All these models are evaluated using the same Neurobench‑provided evaluation pipeline [11] and the same test sets as our method.
> > * For baselines that have published results, we also reproduce their performance to verify consistency with the reported result.
> >
> > **Based on these precautions, we believe that our comparison protocol is reasonably fair.**
> >
> > We also want to clarify how we position our performance claims. We emphasize that our model achieves **state‑of‑the‑art performance among SNN‑based approaches**, but we do not claim that it universally surpasses large ANN models in terms of decoding accuracy.
> > For example, on the NLB_RTT generalization task, our method outperforms POYO‑1 (the SOTA cross‑condition pretraining model), whereas on the more challenging NLB_MAZE task (Table 4), its decoding accuracy is lower than POYO‑1.
> > Since MAZE is the most complex task in our evaluation suite, this is consistent with the expectation that **large‑scale ANN models are more likely to achieve higher decoding accuracy on sufficiently complex tasks**.
> >
> > What we would like to stress is that, under **comparable decoding accuracy**, our method provides **a substantial reduction in the computational cost (implying lower energy consumption) and an extremely lightweight model size (implying better on‑chip memory feasibility)**. These properties are particularly important for practical iBMI applications, where power and memory budgets are strict constraints.
> >
> > >**Table 4.** Performance ($R^2\times100$) on NLB dataset.
> > >
> > >| Model | Variant        | NLB_MAZE (Monkey Je) | NLB_RTT (Monkey I) |
> > >|-------|----------------|----------------------|--------------------|
> > >| ANN   | GRU            | 88.87                | 59.51              |
> > >...
> > >|       | POYO-SS        | 94.70                | 68.50              |
> > >|       | POYO-1         | **94.82**            | 73.78              |
> > >...
> > >| Ours  | RAT SNN-SS     | 90.76±0.30           | 76.34±0.54         |
> > >...
> > >|       | RAT SNN-CC     | 91.16±0.19           | **78.70±0.66**     |
> >
> >
> >
> >
> >
> > ## **Question 2**
> > We appreciate your careful reading of **Eq. (1): $ H[t] = \alpha V_{\text{mem}}[t - 1] + V_{\text{syn}}[t-1] $**. You are correct that this equation differs from the most commonly used discrete‑time LIF update found in many SNN libraries.
> >
> > Our implementation is based on the **Stork** framework developed at the Friedrich Miescher Institute for Biomedical Research [6]. In Stork, there is an explicit **one‑time‑step delay between the synaptic state and the membrane potential**: the membrane potential at time step $t$ is updated from the membrane potential and synaptic state at time step $t-1$, instead of using the synaptic state already updated at time $t$.
> >
> > This design aims to better **capture synaptic delay observed in biological neurons** and **facilitates asynchronous computation on neuromorphic hardware**.
> > Several recent works have built on Stork with the same LIF formulation [9,10].
> >
> >
> > ## **Question 3**
> > We appreciate your comment that **the term “re‑parameterized channel attention” is confusing**. In the revised manuscript, we have replaced it with **“CST re‑parametrized attention”** throughout the text to avoid this ambiguity.
> >
> > We would also like to clarify that **the original use of the term “channel attention” was motivated by the iBMI setting**: CST signals are recorded from microelectrode arrays, each typically containing 96 microelectrodes, i.e., 96 “channels.”
> > Our method is mainly used to compute attention relationships among the 96/192 channels (corresponding to one or two microelectrode arrays, depending on the subject’s implantation), which is why we originally referred to it as “channel attention.”
> >
> > However, as you correctly summarized, this is conceptually different from the **classical channel attention** used in computer vision, such as SE‑Net, where a squeeze‑and‑excitation operation rescales feature‑map channels to reduce redundancy and enhance salient features. Our module instead performs **token‑level self‑attention across electrode channels** using Q/K/V and spike‑driven attention, rather than re‑scaling feature channels via a global pooling and excitation scheme.

---

> > > ### Author Response · Authors · 2025-11-20
> > > **Response to Reviewer tbCB on Question 4-5**
> > >
> > > ## **Question 4**
> > > We appreciate your comment and agree that **our use of the term “global BN” was unclear and potentially misleading**. This question is very helpful for us to improve the presentation.
> > >
> > > What we intended to describe is the following. After computing Q/K/V, we apply BN **separately for each attention head** on its corresponding Q/K/V (i.e.,  BN1D with feature = 1 per head). Each head has its own BN and is normalized independently before the attention computation. In contrast, for the output of the other Rep‑FC layers after the attention operation, we apply **element‑wise BN** to stabilize training.
> > >
> > > As you correctly pointed out, calling this “global BN” is confusing and does not reflect any real novelty. The first strategy is in fact closer to **head‑wise BN** and is a fairly common practice in related architectures. Therefore, in the revised manuscript we have **removed the ambiguous “global BN” wording from the main text** and instead clearly describe our two BN strategies in the Appendix (Section B.5):
> > >
> > > > We apply two BN strategies in the CST re‑parameterized attention block. First, for the Q/K/V computation we use head‑wise BN (i.e., a separate BN for each attention head) to preserve each head’s independence before the attention calculation. Second, after the attention operation we apply element‑wise BN to the output of other Rep‑FC to ensure training stability.
> > >
> > >
> > > ## **Question 5**
> > > Yes, in our paper **“cross-condition” denotes a broader notion that encompasses cross-subject, cross-session, and other sources of variability**.
> > >
> > > We already attempted to convey this in the Introduction and Experiments sections, for example:
> > > >“…to train foundation neural decoding models that generalize across varied conditions including sessions, subjects, and tasks.”
> > >
> > > >“…enables effective integration of neural variability across conditions, including across time, subjects, tasks, experimental environments.”
> > >
> > > To avoid any possible misunderstanding, in the revised manuscript we further clarify this definition in Section 3.3:
> > > > Depending on the scale of the training set, this dataset was organized into three hierarchical levels: **cross-condition** (encompassing multiple subjects and tasks), **cross-session** (within-subject, across sessions), and **single-session**.

---

> > > > ### Author Response · Authors · 2025-11-20
> > > > **Response to Reviewer tbCB on Question 6**
> > > >
> > > > ## **Question 6**
> > > > - What are the model parameters and the time step length used in the SNN in this paper?
> > > > - Additionally, how is the time step defined or segmented for the CST signals?
> > > >
> > > > For the **parameters scale** of the unpruned RAT‑SNN, we already provide a comparison in Section 4.2:
> > > > >Notably, RAT SNN delivers these results with 600K parameters,  significantly fewer than bigRSNN (1.2M) and POYO-SS (1.9M).
> > > >
> > > > Due to space limitations, the **detailed hyperparameters** are listed in the appendix: **Table 8** summarizes the architectural hyperparameters of the SNN models, and **Table 9** lists the neuron‑level hyperparameters.
> > > >
> > > > >**Table.** Hyperparameter settings for RAT SNN.
> > > > >
> > > > >| Parameter                          | Value   |
> > > > >|------------------------------------|--------:|
> > > > >| Input size                         | 192     |
> > > > >| Attention head num ($n_{\text{head}}$) | 4   |
> > > > >| Dimension of qkv ($n$, $d$)        | (192, 1) |
> > > > >| Rep-conv kernel size               | 3       |
> > > > >| Stride                             | 1       |
> > > > >| Dilation                           | 1       |
> > > > >| Bias                               | False   |
> > > > >| MLP size                           | (512, 1) |
> > > > >| Readout head num                   | 5       |
> > > > >| Learning rate                      | 1.00E-02 |
> > > > >
> > > > >**Table.** Time constants and training parameters of LIF and LI neurons.
> > > > >
> > > > >
> > > > >| Parameter                                   | Value    |
> > > > >|---------------------------------------------|---------:|
> > > > >| Synaptic time constant (s)                  | 1.00E-02 |
> > > > >| LIF membrane time constant (s)              | 2.00E-02 |
> > > > >| LI membrane time constant (s)               | 5.00E-02 |
> > > > >| Time constants heterogeneous                | True     |
> > > > >| Time constants trainable                    | True     |
> > > > >| L2 strength of ALB $\lambda_{\mathrm{ALB}}$ | 100      |
> > > > >| L2 threshold of ALB $\phi_{\mathrm{ALB}}$   | 1.00E-03 |
> > > > >| L2 strength of AUB $\lambda_{\mathrm{AUB}}$ | 0.01     |
> > > > >| L2 threshold of AUB $\phi_{\mathrm{AUB}}$   | 10       |
> > > > >| Dropout rate                                | 0.3      |
> > > >
> > > > Regarding the time step and segmentation of the CST signals, we follow the NeuroBench standard [11], where the SNN time step is aligned with the temporal resolution of the CST data （**4 ms**）. This is described in the second paragraph of the “Datasets” section in appendix:
> > > > >All samples from the datasets were uniformly downsampled at 4 ms intervals, with z-score normalization applied per-session basis. The model was trained and validated using overlapping 2-second samples, while testing employed continuous long sequences to simulate real-time processing.
> > > >
> > > > At each time step, RAT‑SNN receives the spike input corresponding to that 4 ms bin and produces the kinematic output for the **same** time step.
> > > >
> > > >
> > > >
> > > >
> > > > ## Summary
> > > >
> > > > We appreciate you again for the detailed review. If you still has any remaining questions or concerns, we would be glad to hear that.
> > > >
> > > >
> > > >
> > > > ****
> > > >
> > > > [1] Zhou Z, Zhu Y, He C, et al. Spikformer: When spiking neural network meets transformer[J]. arXiv preprint arXiv:2209.15425, 2022.
> > > >
> > > > [2] Zhang J, Tang L, Yu Z, et al. Spike transformer: Monocular depth estimation for spiking camera[C]//European Conference on Computer Vision. Cham: Springer Nature Switzerland, 2022: 34-52.
> > > >
> > > > [3] Yao M, Hu J K, Hu T, et al. Spike-driven transformer v2: Meta spiking neural network architecture inspiring the design of next-generation neuromorphic chips[J]. arXiv preprint arXiv:2404.03663, 2024.
> > > >
> > > > [4] Yao M, Hu J, Zhou Z, et al. Spike-driven transformer[J]. Advances in neural information processing systems, 2023, 36: 64043-64058.
> > > >
> > > > [5] Fang W, Yu Z, Chen Y, et al. Deep residual learning in spiking neural networks[J]. Advances in Neural Information Processing Systems, 2021, 34: 21056-21069.
> > > >
> > > > [6] Rossbroich J, Gygax J, Zenke F. Fluctuation-driven initialization for spiking neural network training[J]. Neuromorphic Computing and Engineering, 2022, 2(4): 044016.
> > > >
> > > > [7] Wang Y, Wang Z, Liu S C. Leveraging recurrent neural networks for predicting motor movements from primate motor cortex neural recordings[C]//2024 IEEE Biomedical Circuits and Systems Conference (BioCAS). IEEE, 2024: 1-5.
> > > >
> > > > [8] Azabou M, Arora V, Ganesh V, et al. A unified, scalable framework for neural population decoding[J]. Advances in Neural Information Processing Systems, 2023, 36: 44937-44956.
> > > >
> > > > [9] Gygax J, Zenke F. Elucidating the theoretical underpinnings of surrogate gradient learning in spiking neural networks[J]. Neural Computation, 2025, 37(5): 886-925.
> > > >
> > > > [10] Halvagal M S, Zenke F. The combination of Hebbian and predictive plasticity learns invariant object representations in deep sensory networks[J]. Nature neuroscience, 2023, 26(11): 1906-1915.
> > > >
> > > > [11] Yik J, Van den Berghe K, den Blanken D, et al. The neurobench framework for benchmarking neuromorphic computing algorithms and systems[J]. Nature communications, 2025, 16(1): 1545.

---

> > > > > ### Author Response · Authors · 2025-11-27
> > > > > **Official Comment by Authors**
> > > > >
> > > > > Dear Reviewer tbCB, We want to express our gratitude for dedicating your valuable time to review our paper meticulously and provide us with your valuable insights and feedbacks, which have significantly improved the quality of our work.
> > > > >
> > > > > As the Author-Review Discussion period is approaching the end, we would like to ensure that we have addressed all the concerns you raised. If you have any remaining questions or issues, we would be very happy to provide further clarification or make any necessary modifications.

---

### Official Review · Reviewer_a8mk · 2025-10-30

**Soundness:** 3
**Presentation:** 2
**Contribution:** 3
**Rating:** 6
**Confidence:** 3

**Summary:**

This paper proposes a novel spiking neural network model for neural decoding across heterogeneous sessions, called RAT SNN. Incorporating a re-parameterized architecture, RAT SNN achieves consistent improvement over leading SNN baselines and comparable with ANN baseline POYO. Overall, with SNN being a light-weight model, I believe this is a valuable addition to the community, hence I am leaning towards acceptance.

**Strengths:**

- Propose a novel SNN architecture including re-parameterized self attention and condition-specific batch normalization
- Proposed model show improved performance (in or out of distribution) over previous SNN models and comparable results as non-SSN transformer, which is known to have better decoding performance than SNN
- Scaling analysis shows that the proposed model can benefit from training over more and diverse datasets, which is a promising trait in the era of large-scale pretraining.

**Weaknesses:**

- The method section is in general difficult to follow, especially for someone not very familiar with SNN. It would make it easy to include some high-level description on what particular layers do, instead of laying out detailed equations and terminology
- Effect on dataset scaling is still not clear.. Despite having more data, cross-condtion training is sometimes worse than cross-subject training and even single sessionm in terms of decoding performance, which warrants further investigation.

**Questions:**

1. In appendix (line 740 page 14), the training and validation setups are mentioned to be overlapping 2-sec samples and continuous long sequence respectively. Were the same setup used for POYO as well? Because POYO by default seems to be working with 1-sec segment, with no length generalization in validation. I believe it is worth mentioning that when discussing POYO’s scores.
2. Could the authors comment on the potential reason cross-condition results are sometimes worse than cross-session or single session? For example, for monkey I and L, there is a ~2% improvement from SS to CS, while CC in monkey I is worse than CS. Considering that from SS to CS, the dataset increase is not that big (1 session vs 3 sessions), whereas CC has much more data. Would it be a data issue (maybe dataset is too diverse) or a model issue?
3. In addition, have the authors tried POYO in CC? It seems to be a bigger improvement from SS to CS for POYO, so it would be interesting to compare dataset scaling capability between ANN and SNN.
4. Being at the core of the architecture, it would be better to formally define re-parameterization-style layer.
5. Pruning is not properly defined in section 3.3, but there are multiple mentions of that in the results table. This could cause confusion.
6. In Table 1, perich dataset is included but I didn’t find the results on that dataset. Could the author point to where in the paper that dataset is evaluated?
7. In line 374 page 7, it is mentioned that RTT was used to test transferability to unseen sessions. How about Maze dataset whose results are also in Table 4? Could the authors comment on the experimental setup on that dataset, based on my understanding it couldn’t be in-distribution.
8. Please note that the provided code link is not valid, as all the files are shown not found
9. In equation (4), please mention specifically which index to sum over with respect to the feed-forward function
10. Please fix the numbering error in appendix

---

> ### Author Response · Authors · 2025-11-20
> **Response to Reviewer a8mk (1/3)**
>
> We thank you for the careful reading of our paper and constructive comments in detail.
>
> ## **Weakness 1 and Question 4:**
> Thank you for this helpful suggestion about **"include some high-level description on what particular layers do"** and **"it would be better to formally define re-parameterization-style layer"**.
>
> We use **"CST re‑parametrized attention block"** to replace the “re‑parameterized channel attention block” term in the revised version, following your and other reviewers’ suggestions.
> And we have **updated "Section 3.2: CST re-parametrized self-attention"** to include additional high-level descriptions of the roles and functionalities of the key structure, while still keeping the essential equations：
> >The main body of the CST attention block is a multi-head self-attention. By leveraging the binary nature of SNN, the computational complexity of the attention operation can be reduced from $O(n^2 d)$ to $O(n d^2)$, as in linear attention, where $n \gg d$. In combination with Recurrent-LIF:
> >$$
>     \text{Attention}_ {s_i} = \text{RLIF}((Q_{s_{i}}K_{s_{i}}^T)V_{s_{i}}) = \text{RLIF}(Q_{s_{i}}(K_{s_{i}}^TV_{s_{i}}))
> >$$
> >where $ Q_{s_i} $, $ K_{s_i} $ and $ V_{s_i} $ respectively represent query(Q), key(K) and value(V) of $ i $-th attention head with spike representation, and $ \text{RLIF}(\cdot) $ denotes the temporal dynamics of Recurrent-LIF.
> >
> >Previous studies have shown that convolution layer (Conv) can effectively enhance the performance of spike-driven attentions [1]. However, considering the discontinuity of electrode channel indices and inter-subject differences, we integrate parallel Conv and FC for Qs/Ks/Vs, where FC captures global features and Conv focuses on local features across CST channels:
> >$$
>     Q_s/K_s/V_s = \text{RLIF}(\text{Rep-Conv}(X_{CST}) + \text{Rep-FC}(X_{CST}))
> >$$
> >where $ X_{CST} $ denotes the input CST spikes at current time step and $ \text{Rep-Conv}(\cdot) $ and $ \text{Rep-FC}(\cdot) $ denote the re-parameterized Conv and FC.
> >
> >Moreover, previous experiments have shown that overly deep SNNs can impair decoding accuracy and real-time efficiency in CST decoding for BCI systems and other applications [6].
> Therefore, we merge the MLP block and the output layer in the attention block from prior spike-driven transformer designs [1-4], reducing the depth from the classic five layers to three; this shallower structure not only helps improve accuracy but also reduces model size and computational operations.
> >
> >To further reduce model depth, we compute Qs/Ks/Vs directly from the input CST spikes and, since they lack membrane potentials, replace the membrane-shortcut (MS) in spike-driven transformers with a spike-element-wise shortcut [5] followed by a linear layer (a ``surrogate shortcut''):
> >$$
>     U_ s = \text{RLIF}(\text{Attention}_ s + \text{Rep-FC}(X_{CST}))
> >$$
> >where $ U_s $ denotes the output spikes of CST re-parametrized attention block at current time step. Surrogate shortcut enables changing the attention output dimension ($\text{MLP Size}>n\times{d}$, Fig.1), providing flexible output dimensionality and improving performance without increasing depth.
>
> This makes the method section more accessible to readers and more clearly highlights our architectural innovations.
>
> ## **Weakness 2 and Question 2:**
> We thank you for raising the important question about **why cross-condition (CC) training does not always yield better performance than cross-session (CS) or single-session (SS) training.**
>
> In our experiments, we indeed observe that, on average, cross-condition pretraining brings clear benefits over cross-session pretraining, but in some specific cases the CC results are worse than the CS results or SS results, despite having substantially more training data.
> We believe this is mainly due to the **extreme variability in neural activity**:
> 1. Large differences in electrode layouts and recording qualities across subjects and laboratory settings,
> 2. Noticeable temporal drift within the same subject over time,
> 3. For some datasets, such as Monkey I, signals were occasionally recorded while cortical electrical stimulation was being applied, which introduces severe distribution shifts in the neural activity.
>
> Even with our stepwise training framework, these strong sources of variability cannot be fully compensated. **Similar phenomena have also been reported in prior work such as POYO [7] and NDT3 [8]**, where large-scale pretraining is not uniformly beneficial across all tasks and conditions.
>
> Therefore, our interpretation is aligned with you: **the main reason is that the increased data diversity also amplifies distributional variability and instability in the neural recordings.** And we plan to explore more robust architectures and training strategies that can better exploit highly heterogeneous cross-condition datasets in future work.

---

> > ### Author Response · Authors · 2025-11-20
> > **Response to Reviewer a8mk (2/3)**
> >
> > ## **Question 1:**
> > Yes, your understanding of POYO’s training setup is correct. For POYO, we also use the default **1-second segments** for both training and validation.
> >
> > In practice, different baseline models adopt their own task-specific training and preprocessing strategies—for example, POYO uses a random offset sampler and unit dropout, while AEGRU employs Gaussian noise augmentation and jitter. **It is difficult to impose a single, unified preprocessing and sampling scheme that would align all baselines at once**.
> >
> > Therefore, for each baseline in our experiments, we **follow the hyperparameters and preprocessing strategies recommended by their official implementations**, including POYO’s 1-second segment setting. This choice is intended to ensure that our reported baseline results faithfully reflect the performance of the respective methods under their standard, author-approved configurations.
> >
> > As your suggestion, we have added a clarification in the revised manuscript in Appendix B.8 (*Baseline Set up*):
> >
> > ## **Question 3:**
> > We agree that **comparing dataset scaling capabilities between ANN and SNN is a very interesting direction**, and we plan to investigate this more systematically in future work.
> >
> > For POYO specifically, we did not run our own CC experiments because the authors did not release CC-pretrained weights (POYO-1, their CC-pretrained model). Instead, **in the NLB setting we report POYO’s performance on MAZE and RTT directly from the original paper [7], including both POYO-SS and POYO-1 (POYO-CC, Table 4).**
> > **As also shown in the POYO paper [7], the POYO-CC does not always outperform POYO-CS or even POYO-SS on seen conditions, although POYO-CS performs better than POYO-SS.**
> >
> > >**Table 4.** Performance ($R^2\times100$) on NLB dataset.
> > >
> > >| Model | Variant        | NLB_MAZE             | NLB_RTT            |
> > >|-------|----------------|----------------------|--------------------|
> > >| ANN   | GRU            | 88.87                | 59.51              |
> > >...
> > >|       | POYO-SS        | 94.70                | 68.50              |
> > >|       | POYO-1         | **94.82**            | 73.78              |
> > >...
> > >| Ours  | RAT SNN-SS     | 90.76±0.30           | 76.34±0.54         |
> > >...
> > >|       | RAT SNN-CC     | 91.16±0.19           | **78.70±0.66**     |
> >
> > POYO remains a very strong ANN baseline for CST decoding and represents the top tier in terms of decoding accuracy. **Our goal is not to claim superiority over POYO in accuracy, but to achieve comparable performance**. For example, on NLB_MAZE, POYO-1 outperforms RAT SNN-CC, even though MAZE is out-of-distribution for POYO-1 but in-distribution for RAT SNN-CC. Since MAZE is the most complex task in our evaluation suite, this is consistent with the expectation that **large‑scale ANN models are more likely to achieve higher decoding accuracy on sufficiently complex tasks**.
> >
> > We would like to emphasize that the main advantages of RAT SNN lie in its **very low computational cost (energy potential advantage)** and **small parameter scale (on-chip memory potential advantage)**. Moreover, RAT SNN produces outputs at every time step given the current spike input, while POYO predicts over fixed-length input segments; combined with the much shallower depth, RAT SNN also has **potential advantages for real-time inference** in iBMI applications, where efficiency and latency are as important as decoding accuracy.
> >
> >
> > ## **Question 5:**
> > We thank you for pointing out that **pruning was not clearly defined**.
> >
> > In the original submission, pruning is described in Supplementary Section B.3 (*Lightweight finetuning and pruning*) and Algorithm 1. To avoid confusion, we have also clarified the description in the main text. In the revised version, Section 3.3 now explicitly states:
> > > Then, we provide an optional iterative prune step (masked the smallest $p$ of weights) and following retrain to further reduce model connection size and computational complexity.
> >
> > This makes the pruning procedure and its role in our pipeline more explicit for the reader.

---

> ### Author Response · Authors · 2025-11-20
> **Response to Reviewer a8mk (3/3)**
>
> ## **Question 6:**
> About **the results on perich dataset**, it is used only in our generalization to unseen conditions experiments. Specifically, it appears in **Fig. 5** of the main paper and **Fig. 15** in the appendix.
>
> Perich is not included in the CC pretraining dataset. Instead, we treat it as an unseen condition and compare fine-tuning RAT SNN-CC on Perich with training RAT SNN-SS (from scratch) on the same dataset. This setup is used to demonstrate that our pretrained model can quickly and effectively adapt to new, unseen conditions.
>
> In the revised manuscript, we have also added, at the end of appendix Section C (*Evaluation details*), the corresponding comparisons for the sessions used in Fig. 5, showing that RAT SNN-CC outperforms RAT SNN-SS both in final decoding accuracy and in convergence speed.
>
> >**Table 12.** Performance comparison on unseen conditions on sessions used in Fig. 5 and Fig. 15
> >| Model        | Metric          | Monkey M        | Monkey T        | Monkey C        |
> >|-------------|-----------------|-----------------|-----------------|-----------------|
> >| RAT SNN-SS  | $$R^2 \times 100$$ | 86.99±0.51      | 74.84±0.32      | 93.21±0.17      |
> >|             | Epoch           | 231.56±36.40    | 304.13±64.91    | 364.75±68.26    |
> >| **RAT SNN-CC** | **$$R^2 \times 100$$** | **87.11±0.15** | **76.29±1.09** | **93.30±0.11** |
> >|             | **Epoch**       | **167.38±48.19**| **106.63±16.65**| **221.13±37.28**|
>
> ## **Question 7:**
> Regarding the MAZE dataset, it is **not used as an unseen condition for RAT SNN-CC**. In our experimental setup, MAZE is **included in the CC pretraining dataset**. We made this choice so that the CC pretraining covers at least three different task types (center-out, random target, and maze). As a consequence, in Table 4 the MAZE results for RAT SNN-CC should be interpreted as **in-distribution**, whereas RTT is used to evaluate transferability to unseen sessions.
>
> We agree that the current ecosystem offers only a limited variety of tasks which constrains the diversity of our pretraining and evaluation.
> In future work, we hope to leverage more diverse datasets and task types for pretraining and evaluation to better assess model generalization.
>
>
> ## **Question 8:**
> We thank you for pointing out that **"code link is not valid"**. We used an anonymous GitHub/4open link during the review process, which sometimes caused an extra (and unreliable) redirection step, leading to the “files not found” issue.
>
> In the revised manuscript, **we have updated the link** to avoid this redirection. The code can now be accessed directly at:
> <https://anonymous.4open.science/r/RAT-SNN-FA7E>
>
> This link remains fully anonymous and therefore does not violate the ICLR double‑blind review policy.
>
> ## **Question 9:**
> For Equation (4), regarding **“which index to sum over with respect to the feed-forward function,”** we have made this explicit in the revised manuscript.
>
> The updated equation is:
> $$
> V_ {\text{syn}}^l[t] = \sum_{i=1}^N f^l_ {op_i}\big(S^{u}_ i[t]\big) + f^l_{\text{Rec}}(S^{l}[t-1]) + \beta^l V_{\text{syn}}^l[t-1],
> $$
> where the summation is over the branch index $i$.
>
> We also clarify this in the text as follows:
> > where $f^l_{op}(\cdot)$ denotes the feedforward connections of upstream spike $S^{u}$ in LIF layer $l$ with $i$ indicate different branches.
>
>
> ## **Question 10:**
> I have corrected the numbering errors in the appendix; thank you for the reminder.
>
>
>
>
>
> ## Summary
>
> We appreciate you again for the detailed review. If you still has any remaining questions or concerns, we would be glad to hear that.
>
>
>
>
> [1] Yao M, Hu J K, Hu T, et al. Spike-driven transformer v2: Meta spiking neural network architecture inspiring the design of next-generation neuromorphic chips[J]. arXiv preprint arXiv:2404.03663, 2024.
>
> [2] Zhang J, Tang L, Yu Z, et al. Spike transformer: Monocular depth estimation for spiking camera[C]//European Conference on Computer Vision. Cham: Springer Nature Switzerland, 2022: 34-52.
>
> [3] Zhou Z, Zhu Y, He C, et al. Spikformer: When spiking neural network meets transformer[J]. arXiv preprint arXiv:2209.15425, 2022.
>
> [4] Yao M, Hu J, Zhou Z, et al. Spike-driven transformer[J]. Advances in neural information processing systems, 2023, 36: 64043-64058.
>
> [5] Fang W, Yu Z, Chen Y, et al. Deep residual learning in spiking neural networks[J]. Advances in Neural Information Processing Systems, 2021, 34: 21056-21069.
>
> [6] Rossbroich J, Gygax J, Zenke F. Fluctuation-driven initialization for spiking neural network training[J]. Neuromorphic Computing and Engineering, 2022, 2(4): 044016.
>
> [7] Azabou M, Arora V, Ganesh V, et al. A unified, scalable framework for neural population decoding[J]. Advances in Neural Information Processing Systems, 2023, 36: 44937-44956.
>
> [8] Ye J, Rizzoglio F, Smoulder A, et al. A Generalist Intracortical Motor Decoder[J]. bioRxiv, 2025.

---

> ### Comment · Reviewer_a8mk · 2025-11-23
>
> I thank the author for detailed responses to my review, as well as the additional experimental results on perich dataset. I believer it improves the overall quality of the manuscripts. I do have several additional comments:
>
> > We believe this is mainly due to the extreme variability in neural activity
>
> I understand the potential variability across sessions, so I am not concerned in terms of any individual session, but rather on a population level, an improved performance in average. This applies for both seen and unseen sessions. For seen sessions significant amount of session underperforming can be a sign of insufficient model capacity, and it would be interesting to see the scaling analysis. Meanwhile, I am more interested in the unseen sessions. Considering that the C05_2025 is just one session, I agree that it can be vastly different from the pretraining dataset, so you might need more unseen sessions from C05 for a more comprehensive understanding. Hence I appreciate the inclusion of perich dataset as unseen sessions, and I think it shows promising results.
>
> In addition, regarding POYO-1, I would like to mention that in terms of finetuning to unseen sessions, cross-condition POYO1 is better than SS counterpart in average, in all the reported table of the original paper. POYO1 with unit identification was indeed slightly worse than SS, but that is a special finetuning strategy where >95% parameters were frozen.
>
> > Regarding the MAZE dataset, it is not used as an unseen condition for RAT SNN-CC. In our experimental setup, MAZE is included in the CC pretraining dataset. We made this choice so that the CC pretraining covers at least three different task types (center-out, random target, and maze). As a consequence, in Table 4 the MAZE results for RAT SNN-CC should be interpreted as in-distribution, whereas RTT is used to evaluate transferability to unseen sessions.
>
> Thank you for the explanation, and I agree with your decisions on more task diversity. In this case, I would recommend seen and unseen session results be put in separate tables. Right now it is confusing to see MC Maze and MC RTT side-by-side in the same table, and I guess it extends to C05_2022 and C05_2025 as well. Maybe the unseen results can be put in one table. But either way they need to be more clearly indicated on the table (or table's title).

---

> > ### Author Response · Authors · 2025-11-27
> > **Response to Reviewer a8mk (1/3)**
> >
> > Thank you for the quick and detailed response to our update.
> > We updated the manuscript again reflecting your additional feedback and our response to your comments is as follows:
> >
> > >For seen sessions significant amount of session underperforming can be a sign of insufficient model capacity, and it would be interesting to see the scaling analysis.
> >
> > Sorry for not clearly presenting our results, we would like to kindly clarify that, **on most seen sessions, RAT SNN-CC (cross-condition pretrained) performs better than RAT SNN-CS (cross-session pretrained， including all available sessions from the same monkey) and RAT SNN-SS (single-session trained)**.
> > Specifically, RAT SNN-CC outperforms RAT SNN-CS on 6 out of 9 test sessions, and their performances are very similar on 2 of the remaining 3 sessions. Moreover, on nearly all sessions both RAT SNN-CC and RAT SNN-CS outperform RAT SNN-SS in decoding accuracy (except Monkey I session 2, Table 1).
> >
> >
> > **Table 1.** Performance comparison on seen sessions. Best and second-best models are in bold and underlined, respectively.
> > | Dataset                | Session   | RAT SNN-SS                  | RAT SNN-CS                           | RAT SNN-CC                           |
> > |------------------------|-----------|-----------------------------|--------------------------------------|--------------------------------------|
> > | Monkey I               | session 1 | 78.06±0.77              | $\underline{79.68±0.29}$           | **80.13±0.32**              |
> > |                        | session 2 | $\underline{63.29±0.99}$  | **64.07±0.61**              | 63.04±1.04                       |
> > |                        | session 3 | 75.33±1.01              | **79.03±0.54**              | $\underline{79.01±0.46}$           |
> > | Monkey L               | session 1 | 66.88±2.43              | **69.12±1.20**              | $\underline{68.99±1.80}$           |
> > |                        | session 2 | 60.87±2.51              | $\underline{62.11±0.80}$           | **66.16±0.30**              |
> > |                        | session 3 | 71.15±0.87              | $\underline{74.68±0.56}$           | **76.05±0.80**              |
> > | Monkey C05\_2022       | session 1 | 84.24±0.28              | $\underline{85.37±0.36}$           | **85.94±0.21**              |
> > |                        | session 2 | 85.29±0.86              | $\underline{87.52±0.38}$           | **87.73±0.17**              |
> > |                        | session 3 | 75.20±0.88              | $\underline{80.13±0.56}$           | **81.53±0.61**              |
> > | Monkey Je (NLB\_MAZE)  |           | 90.76±0.30              | ----                                 | **91.16±0.19**              |
> >
> > Due to the relatively poor performance of RAT SNN-CC on Monkey I session 2, its average performance on Monkey I is lower than that of RAT SNN-CS. To further clarify this, we additionally report results from three additional sessions of Monkey I (session 4-6, Table 2). Conducted on these extra sessions, we can see that **both RAT SNN-CC and RAT SNN-CS consistently outperform RAT SNN-SS in decoding accuracy, and RAT SNN-CC achieves the best average performance.**
> >
> > **Table 2.** Performance comparison on additional Monkey I sessions.
> > | Dataset  | Session   | RAT SNN-SS             | RAT SNN-CS                     | RAT SNN-CC                     |
> > |----------|-----------|------------------------|--------------------------------|--------------------------------|
> > | Monkey I | session 1 | 78.06±0.77         | $\underline{79.68±0.29}$    | **80.13±0.32**       |
> > |          | session 2 | $\underline{63.29±0.99}$ | **64.07±0.61**   | 63.04±1.04                |
> > |          | session 3 | 75.33±1.01         | **79.03±0.54**       | $\underline{79.01±0.46}$    |
> > |          | session 4 | 75.42±0.40         | $\underline{76.84±0.45}$    | **76.91±0.73**       |
> > |          | session 5 | 82.19±0.62         | **84.26±0.46**       | $\underline{83.90±0.89}$    |
> > |          | session 6 | 82.53±0.69         | $\underline{83.73±0.48}$    | **84.88±0.87**       |
> > |          | Avg       | 76.10±0.23         | $\underline{77.93±0.11}$    | **77.98±0.28**       |
> >
> >
> > Importantly, consistent with your suggestion, we also believe that scaling up to increase the capacity of RAT SNN is very likely to further strengthen the advantage of RAT SNN-CC over RAT SNN-CS.
> > However, this goes against our original goal, as we aim to **keep our model lightweight in terms of both the number of parameters and computational cost** which is as important as decoding accuracy in iBMI applications. Considering the current allocation of computational resources in our lab and the limited time available for the rebuttal, we may not be able to include additional scaling-up comparison experiments before the end of the rebuttal period, but we would be happy to add the subsequent results to the camera-ready version or to GitHub for further discussion.

---

> > > ### Author Response · Authors · 2025-11-27
> > > **Response to Reviewer a8mk (2/3)**
> > >
> > > >Considering that the C05_2025 is just one session, I agree that it can be vastly different from the pretraining dataset, so you might need more unseen sessions from C05 for a more comprehensive understanding.
> > >
> > > We fully agree with your comment and have additionally included one more C05_2025 session for analysis. The results on this extra session (Table 3) further confirm that, **on most unseen sessions from seen animal, RAT SNN-CC and RAT SNN-CS not only outperform RAT SNN-SS in decoding accuracy but also significantly accelerate convergence, and overall, RAT SNN-CC achieves slightly better performance than RAT SNN-CS.**
> > > Although in Monkey C05_2025 session 1 RAT SNN-SS is better than RAT SNN-CS and RAT SNN-CC, its performance on the validation set is unstable and inferior to RAT SNN-CS and RAT SNN-CC (Fig. 14 in our paper).
> > >
> > > It is worth noting that, under unseen conditions, **RAT SNN-CC outperforms RAT SNN-SS in both decoding accuracy and convergence speed on all sessions from the three unseen monkeys in the unseen dataset (Fig. 5 and Fig. 15 in our paper)**.
> > >
> > > **Table 3.** Performance comparison on unseen sessions.
> > > |              |               | Monkey C05\_2025        |                         | Monkey I                 |
> > > |--------------|---------------|-------------------------|-------------------------|--------------------------|
> > > |              |               | session 1               | session 2               | NLB\_RTT                 |
> > > | RAT SNN-SS   | $R^2\times100$| **69.17±0.75** | 64.25±1.01          | 76.34±0.54           |
> > > |              | Epoch         | 46.4000±10.1882     | 48.4000±6.5422      | 171.6000±74.8184     |
> > > | RAT SNN-CS   | $R^2\times100$| $\underline{68.91±1.03}$ | $\underline{64.64±1.21}$ | $\underline{78.55±0.19}$ |
> > > |              | Epoch         | 28.5000±5.0662      | 20.2500±2.0616      | 88.2500±39.3732      |
> > > | RAT SNN-CC   | $R^2\times100$| 68.21±1.00          | **64.87±1.59** | **78.70±0.66**  |
> > > |              | Epoch         | 28.7500±4.9244      | 21.7500±1.5000      | 126.7500±20.1722     |
> > >
> > > Due to the inclusion of Monkey C05_2025 session 2, we have correspondingly updated the Monkey C05_2025 results in our ablation studies in the revised manuscript for consistency, and we have uploaded this session to GitHub to facilitate reproduction of our experimental results.
> > >
> > > Moreover, we would like to clarify that the electrodes for Monkey C05 were implanted in 2019, and by 2025 the neural signal quality had degraded, so we were only able to find one additional session of sufficient quality so far; all the other sessions had too few effective channels (<7 of 96) or too low overall firing rates (<1 Hz).
> > > In future work, we plan to include more data from Monkey C05 (Monkey C05_2022–2024) in our analysis, and we believe that as the dataset grows, our experimental conclusions will become even more robust.
> > >
> > > **Above all, we would like to emphasize that, in line with your concern about performance “on a population level,” both RAT SNN-CC and RAT SNN-CS achieve substantial improvements over RAT SNN-SS on both seen and unseen sessions from seen subjects. Although RAT SNN-CC shows only a slight advantage over RAT SNN-CS, it is important to note that RAT SNN-CS is trained separately for each individual monkey, whereas RAT SNN-CC is explicitly designed to generalize to unseen subjects and behavioral tasks.**

---

> ### Author Response · Authors · 2025-11-27
> **Response to Reviewer a8mk (3/3)**
>
> >In addition, regarding POYO-1, I would like to mention that in terms of finetuning to unseen sessions, cross-condition POYO1 is better than SS counterpart in average, in all the reported table of the original paper. POYO1 with unit identification was indeed slightly worse than SS, but that is a special finetuning strategy where >95% parameters were frozen.
>
> We fully agree with your comment on POYO and appreciate your deep familiarity with its results—this insight has greatly facilitated our discussion.
> POYO presents two variants of pretrained models: POYO-1, which is trained on multiple datasets (closely corresponding to our "-CC" setting), and POYO-mp, which is trained on a single dataset (intermediate between our "-CS" and "-CC" settings).
> According to Table 2 in their paper, when generalizing to seen dataset, POYO-1 does not consistently outperform POYO-mp (Monkey C-CO and Monkey T-CO). For the average of two sessions of Monkey C-CO, its performance (0.9683 ± 0.0118) is even nearly identical to that of POYO-[Single-session] (0.9682 ± 0.0111).
> Despite its strong performance on unseen dataset generalization, this observation further underscores the substantial variability in neural activity across distinct datasets, and suggests that models trained on large-scale data (such as POYO-1) may not always outperform those trained on smaller-scale datasets (such as POYO-mp) under seen conditions.
>
> Our experimental results exhibit a similar phenomenon: the performance of RAT SNN-CC is not always better than RAT SNN-CS, or even RAT SNN-SS. A reasonable explanation is that **models with cross-dataset pretraining (such as RAT SNN-CC or POYO-1), while improving overall generalization to unseen datasets, may not show a clear advantage under seen datasets compared to models pretrained specifically for that datasets (such as RAT SNN-CS or POYO-mp)**.
>
> >**Table 2 [1]**: Behavioral decoding results across neural recordings from four nonhuman primates performing two different tasks.
> >|                      | Same animal, New day|New animal         |                    |
> >|----------------------|---------------------|-------------------|--------------------|
> >| Method               | Monkey C - CO (2)   |Monkey T - CO (6)  | Monkey T - RT (6)  |
> >...
> >| POYO-[Single-session]| 0.9682 ± 0.0111     |0.9194 ± 0.0185    | 0.7800 ± 0.0702    |
> >| POYO-mp + Unit ID    | 0.9675 ± 0.0079     |0.9012 ± 0.0271    | 0.7759 ± 0.0471    |
> >| POYO-mp + Finetune   | **0.9708 ± 0.0116** |**0.9379 ± 0.0193**| 0.8105 ± 0.0561    |
> >| POYO-1 + Unit ID     | 0.9677 ± 0.0096     |0.9028 ± 0.0248    | 0.7788 ± 0.0548    |
> >| POYO-1 + Finetune    | 0.9683 ± 0.0118     |0.9364 ± 0.0132    | **0.8145 ± 0.0496**|
>
> >Maybe the unseen results can be put in one table. But either way they need to be more clearly indicated on the table (or table's title).
>
> We appreciate your suggestion. To enhance clarity, we have marked the generalization (unseen session) results with a superscript “+” in all tables to clearly distinguish them from the “from-scratch” training results. As an example:
>
> **Table 5.** Performance ($R^2\times100$) on Monkey C05. “$^+$” indicates generalization results on unseen sessions.
>
> | Model          |               | C05_2022           | C05_2025              |
> |----------------|---------------|--------------------|-----------------------|
> | ANN            | AEGRU         | $76.05±0.43$         | $52.39±0.81$            |
> |                | POYO-SS       | $77.05±0.80$         | $64.93±2.12$            |
> | SNN            | bigRSNN-SS    | $81.72±0.35$         | $62.15±0.63$            |
> |                | bigRSNN-CS    | $83.56±0.54$         | $66.45±0.28^+$        |
> | Ours           | RAT SNN-SS    | $81.58±0.42$         | $66.71±0.48$        |
> |                | RAT SNN-CS    | $\underline{84.34±0.19}$       | $\underline{66.77±1.11}^+$        |
> |                | RAT SNN-CC    | $\textbf{85.07±0.21}$   | $66.54±0.74^+$        |
> |                | RAT SNN-CC-P  | $83.88±0.11$         | $\textbf{67.08±1.36}^+$      |
>
> We have also added merged tables summarizing the unseen-session results in the appendix of the revised manuscript for easier reference (Table 3 above is one example).
>
> ## Summary
>
> We appreciate you again for the quick and detailed response to our update.
>
> [1] Azabou M, Arora V, Ganesh V, et al. A unified, scalable framework for neural population decoding[J]. Advances in Neural Information Processing Systems, 2023, 36: 44937-44956.

---

> > ### Comment · Reviewer_a8mk · 2025-11-27
> >
> > I thank the author for the continued engagement on the discussion and for conducting additional experiments. I believe the new results on the additional sessions consolidate the claims of the paper. I believe my concerns have been addressed and I will change my score. I thank the authors for their effort during the rebuttal period.

---

> > > ### Author Response · Authors · 2025-11-28
> > > **Response to Reviewer a8mk**
> > >
> > > Thank you very much for your positive feedback and for taking the time to review our work so thoroughly. We greatly appreciate your recognition of our additional experiments and your willingness to update your score. Your constructive comments have been invaluable in improving our paper. If you have any further suggestions or questions, please feel free to let us know. Thank you again for your support during the review process!

---

### Official Review · Reviewer_XSu9 · 2025-10-31

**Soundness:** 3
**Presentation:** 2
**Contribution:** 3
**Rating:** 8
**Confidence:** 2

**Summary:**

The authors build on existing Spiking neural networks architecture using attention heads by introducing structural reparametrization in the self-attention blocks. The reparametrization comes with free batch normalisation, and the authors assign to each condition (ie. behaviour task) an independent batch normalisation during pre-training. The resulting model is then put to the bench using a corpus of extracellular neurophysiology recordings in non-human primate performing reaching tasks.
The RAT-SNN model overperform previous SNNs and matches ANNs, at a lower computational expense (pruned version of 150k parameters compared to ~2M for ANNs). Furthermore, the pre-trained model is shown to systematically converge in fewer iterations than zero-shot attempts.

**Strengths:**

- Motivation is clear, SNN could enable generalizable models and be a lower power alternative than conventional ANNs for BMI applications.
- Performance on neural decoding is comparable to ANNs for this experiment, with an ablation study showing the reparametrization having a 0.15 increase on R2.
- The experiment follows a proposed framework for neuromorphic modeling benchmarking for model for baseline comparisons where applicable and for computational (Neurobench)

**Weaknesses:**

The experimental protocol is narrow in the sense that tasks are all very similar, using the same recording modalities in the same cortical areas of primates. One hypothesis could be that ANNs will generalize better than SNNs when the breadth of datasets will expand. Here maybe the experimental protocol is a bit too easy, without enough "unseen" subjects/tasks benchmarking. Even if ANNs were to take a strong scoring lead in such a case compared to SNNs, this wouldn't reduce the appeal of the presented method for real-time applications.
- I struggled to follow the experimental protocol and understand the terms single-session, cross-session, cross-condition: in all the performances of table 2 and 3 the pre-training always seems to include all datasets, while truly "unseen" benchmarks are only introduced in figure 5. Maybe a visual or a table recap of what is used in each training phase and in test would be more clear.
- Private dataset is... private, making any reproducibility or competition attempt impossible at the moment.


## Minor comments
- typo line 961: we employs
- the github repository is unaccessible

**Questions:**

- What experimental benchmark would make SNNs fail compared to ANNs ? Is there a more drastic protocol where pre-trained models are shown a completely unseen Maze task in an unseen subject where the ANNs take a strong lead ? Answering this question would make for a stronger paper.

---

> ### Author Response · Authors · 2025-11-20
> **Response to Reviewer XSu9 on weakness 1**
>
> We are grateful for your careful review of our work and for the insightful, constructive feedback.
>
> ## **Weakness 1:**
> >**The experimental protocol is narrow** in the sense that tasks are all very similar, using the same recording modalities in the same cortical areas of primates. One hypothesis could be that ANNs will generalize better than SNNs when the breadth of datasets will expand. **Here maybe the experimental protocol is a bit too easy**, without enough "unseen" subjects/tasks benchmarking. Even if ANNs were to take a strong scoring lead in such a case compared to SNNs, this wouldn't reduce the appeal of the presented method for real-time applications.
>
> We thank you for highlighting the limited diversity of our current experimental protocol and for recognizing the appeal of our method for real-time applications.
>
> We agree that the present set of tasks is relatively “narrow/easy”. This is mainly due to the current scarcity of publicly available datasets on non-human primate CST recordings during forelimb movements, which we acknowledge as an important limitation of this work. As large-scale cortical recording technologies continue to advance, we expect more diverse datasets to become publicly available, enabling a broader and more systematic evaluation of iBMI decoders (including both SNNs and ANNs).
>
> From the perspective of future extensions, incorporating cross-species cortical spike data (e.g., from rodents or humans) into the pre-training corpus is a promising direction. Although the motor cortices of different species have structural differences, prior work has shown that cross-species pre-training can be feasible and beneficial under certain conditions [1].
>
> In addition, we plan to explore larger-scale unsupervised/self-supervised pre-training that involves more brain areas and tasks—for example, using visual cortical recordings—in order to enrich exploitable dataset.

---

> > ### Author Response · Authors · 2025-11-20
> > **Response to Reviewer XSu9 on weakness 2**
> >
> > ## **Weakness 2:**
> > >I struggled to follow **the experimental protocol** and understand **the terms single-session, cross-session, cross-condition**: in all the performances of table 2 and 3 the pre-training always seems to include all datasets, while truly "unseen" benchmarks are only introduced in figure 5. **Maybe a visual or a table recap of what is used in each training phase and in test would be more clear**.
> >
> > We thank the reviewer for pointing out that our description of the single-session, cross-session, and cross-condition protocols was not sufficiently clear.
> >
> > We summarize the data used at each training stage in Table 1 of the main text and Figure 9 of the appendix, and Figures 10–12 further specify the data used in each pre-training stage. In summary:
> >
> > **Cross-condition (CC):** In our paper CC denotes a broader notion that encompasses cross-subject, cross-task, cross-session, and other sources of variability.The cross-condition dataset includes NHP, NLB_MAZE, and the private dataset (C05_2022).
> >
> > **Cross-session (CS):** In our paper, CS data collected from the same subject across different time periods (including multiple sessions). The cross-session datasets respectively include NHP and the private dataset (C05_2022).
> >
> > **Single-session (SS):** In the datasets used in our work, each session corresponds to one complete behavioral recording from a single animal within a single, continuous experimental setup. Typically, each day’s experiment contains only 1–2 sessions, and each session consists of multiple trials, i.e., repeated executions of the task by the animal.
> > - Dataset segmentation schematic: we partition the data at the session level into training, validation, and test sets (Fig.8).
> >
> > We evaluate decoding performance on all datasets, and further use NLB_RTT and the private dataset (C05_2025) and the Perich dataset to assess the model’s generalization ability.
> >
> > To avoid any possible misunderstanding, in the revised manuscript we further clarify this definition in Section 3.3:
> > > Depending on the scale of the training set, this dataset was organized into three hierarchical levels: **cross-condition** (encompassing multiple subjects and tasks), **cross-session** (within-subject, across sessions), and **single-session**.
> >
> > >**Table 1.** Datasets used in this work. “\*” means used in the cross-condition dataset, while “+” means used in the cross-session dataset. RTT: Random Target Task, CO: Center-Out.
> > >
> > >| Study                   | Subject    | Brain Area | Channels | Task              | Sessions |
> > >|-------------------------|------------|-----------|----------|-------------------|----------|
> > >| NHP [1]\*\+             | Monkey I, L    | M1, S1    | 96/192   | RTT               | 37       |
> > >| NLB_MAZE [2]\*          | Monkey Je      | M1, PMd   | 192      | MAZE              | 1        |
> > >| NLB_RTT [2]             | Monkey I       | M1        | 96       | RTT               | 1        |
> > >| Perich dataset [3]      | Monkey M, J, C, T | M1, PMd | 192      | RTT-joystick, CO  | 53       |
> > >| Private dataset [4]\*\+ | Monkey C05     | M1        | 96       | CO                | 11       |
> >
> > The current ecosystem offers only a limited variety of tasks, which constrains the diversity of our pretraining and evaluation. In future work, we hope to leverage more diverse datasets and task types for pretraining and evaluation to better assess model generalization.
> >
> >
> > **For "unseen" benchmarks:**
> >
> > Table 3 and Table 4 present the results on the “unseen sessions”, namely “C05_2025” and “NLB_RTT”. These two sessions are not included in any of our pre-training datasets. They are used exclusively to evaluate the generalization performance of our method on unseen sessions, i.e., its ability to cope with neural temporal drift.
> >
> >
> > Figure 5 illustrates the generalization ability of our model to an unseen condition, and is mainly used to demonstrate the benefit of our pre-training strategy over training from scratch.
> > In the revised manuscript, we have also added, at the end of appendix Section C (*Evaluation details*), the corresponding comparisons for the sessions used in Fig. 5, showing that RAT SNN-CC outperforms RAT SNN-SS both in final decoding accuracy and in convergence speed.
> >
> > >**Table 12.** Performance comparison on unseen conditions on sessions used in Fig. 5 and Fig. 15
> > >| Model        | Metric          | Monkey M        | Monkey T        | Monkey C        |
> > >|-------------|-----------------|-----------------|-----------------|-----------------|
> > >| RAT SNN-SS  | $R^2 \times 100$ | 86.99±0.51      | 74.84±0.32      | 93.21±0.17      |
> > >|             | Epoch           | 231.56±36.40    | 304.13±64.91    | 364.75±68.26    |
> > >| **RAT SNN-CC** | **$R^2 \times 100$** | **87.11±0.15** | **76.29±1.09** | **93.30±0.11** |
> > >|             | **Epoch**       | **167.38±48.19**| **106.63±16.65**| **221.13±37.28**|

---

> > > ### Author Response · Authors · 2025-11-20
> > > **Response to Reviewer XSu9 on Questions**
> > >
> > > ## **Questions**
> > > >**What experimental benchmark would make SNNs fail compared to ANNs ?**
> > > **Is there a more drastic protocol where pre-trained models are shown a completely unseen Maze task in an unseen subject where the ANNs take a strong lead ?** Answering this question would make for a stronger paper.
> > >
> > > Your hypothesis is very plausible, and we also believe that **under a more stringent protocol, large-scale ANNs would likely take a strong lead**. In our work, however, the MAZE dataset is included in the pre-training corpus of RAT SNN-CC. We designed it this way so that the CC pre-training dataset would contain at least three different tasks (center-out, random target, and maze). As a result, in our generalization experiments, the MAZE dataset is already in-distribution for RAT SNN-CC. Therefore, directly comparing RAT SNN-CC with POYO (the ANN SOTA model) on MAZE would be unfair to POYO, even though POYO actually achieves higher decoding accuracy than RAT SNN-CC on this task.
> > >
> > > Therefore, based on RAT SNN-CC-mini (constructed in Appendix C, *Cross-condition pretraining effectiveness*, using a smaller pre-training dataset with a data scale comparable to NLB_MAZE but excluding NLB_MAZE itself), we additionally fine-tuned RAT SNN-CC-mini on NLB_MAZE as a supplementary experiment.
> > > The fine-tuning results show that RAT SNN-CC-mini achieves a decoding accuracy of 0.9101 on the NLB_MAZE task, which lies between RAT SNN-SS and RAT SNN-CC, and is lower than POYO-1’s 0.9482, consistent with our expectation.
> > >
> > > Since the MAZE task is the most complex task in our evaluation framework, while on the simpler NLB_RTT task RAT SNN-CC attains higher decoding accuracy than POYO-1, we speculate that **for more complex tasks, ANN models are more likely to achieve higher decoding accuracy**.
> > >
> > > **Table.** Performance ($R^2\times100$) on NLB dataset.
> > >
> > > |  | Model        | NLB_MAZE (Monkey Je) |
> > > |-------|----------------|----------------------|
> > > |       | POYO-SS        | 94.70                |
> > > |       | **POYO-1**         | **94.82**            |
> > > | Ours  | RAT SNN-SS     | 90.76±0.30           |
> > > |       | **RAT SNN-CC-mini**| **91.01±0.50**           |
> > > |       | RAT SNN-CC     | 91.16±0.19           |
> > >
> > > **Discussion**
> > >
> > > Because SNNs transmit information in a discrete spiking space and require surrogate gradients for learning, they often become increasingly disadvantaged compared to ANNs as model size grows.
> > > This phenomenon has been observed in many neuromorphic benchmarks, such as the Spiking Heidelberg Digits and DVS-Gesture tasks [6].
> > >
> > > We also observed this phenomenon in our previous experiments: when using a simple fully connected SNN architecture for CST decoding, increasing the number of hidden layers did not improve decoding performance; instead, it actually led to a degradation in accuracy:
> > >
> > > >In the CST decoding task with pretraining (to minimize the adverse effects of limited dataset size on deep models), we compared SNNs with 1–3 hidden layers and found that, when using the simplest fully connected RSNN architecture, increasing the number of hidden layers not only fails to improve decoding performance but in fact leads to a degradation.
> > > >
> > > >**Table. Performance comparison ($R^2\times100$) between 1–3 hidden layers SNN.**
> > > >These SNN model are tested on Monkey I and Monkey L with random seed 1-5, the >hidden LIF layer size is 64.
> > > >| Model       | Monkey I & L Avg. |
> > > >|------------|--------------------|
> > > >| SNN-1hidden | 57.68±0.93       |
> > > >| SNN-2hidden | 54.77±1.44       |
> > > >| SNN-3hidden | 49.21±0.84       |
> > >
> > > Therefore, compared with SNNs, ANN models are more likely to benefit from increased model size and complexity, and thus have greater potential when fitting more complex tasks.
> > >
> > > Overall, although ANNs may have considerable advantages in terms of achievable decoding accuracy, as you pointed out in your weakness section, **our method is primarily motivated by its applicability to real-time iBMI systems, including advantages in real-time performance, computational cost, and model size**.

---

> > > > ### Author Response · Authors · 2025-11-20
> > > > **Response to Reviewer XSu9 on weakness 3**
> > > >
> > > > ## **Weakness 3:**
> > > > >Private dataset is... private, making any reproducibility or competition attempt impossible at the moment.
> > > >
> > > > Regarding the comment on the “Private dataset”, we have released the data of the C05_2025 session together with the project code to enable reproduction of our results.
> > > >
> > > > In addition, the remaining data will be made publicly available after publication of the paper to further support research in this field.
> > > >
> > > > ## **Minor comments**
> > > > >typo line 961: we employs
> > > > >
> > > > >the github repository is unaccessible
> > > >
> > > > Thank you for pointing this out. We have corrected the spelling error in the manuscript and fixed the issue of the anonymous GitHub link.
> > > >
> > > > ## Summary
> > > >
> > > > We appreciate you again for the detailed review. If you still has any remaining questions or concerns, we would be glad to hear that.
> > > >
> > > > [1] Ye J, Rizzoglio F, Smoulder A, et al. A Generalist Intracortical Motor Decoder[J]. bioRxiv, 2025.
> > > >
> > > > [2] Joseph E. O’Doherty, Mariana M. B. Cardoso, Joseph G. Makin, and Philip N. Sabes. Nonhuman
> > > > primate reaching with multichannel sensorimotor cortex electrophysiology, May 2017.
> > > >
> > > > [3] Felix Pei, Joel Ye, David Zoltowski, Anqi Wu, Raeed Chowdhury, Hansem Sohn, Joseph O’Doherty,
> > > > Krishna V. Shenoy, Matthew Kaufman, Mark Churchland, Mehrdad Jazayeri, Lee Miller,
> > > > Jonathan Pillow, Il Memming Park, Eva Dyer, and Chethan Pandarinath. Neural latents bench-
> > > > mark ‘21: Evaluating latent variable models of neural population activity. Proceedings of the
> > > > Neural Information Processing Systems Track on Datasets and Benchmarks, 1, December 2021.
> > > >
> > > > [4] Eva L. Dyer, Mohammad Gheshlaghi Azar, Matthew G. Perich, Hugo L. Fernandes, Stephanie
> > > > Naufel, Lee E. Miller, and Konrad P. K¨ording. A cryptography-based approach for movement
> > > > decoding. Nature Biomedical Engineering, 1(12):967–976, December 2017. ISSN 2157-846X.
> > > >
> > > > [5] Tengjun Liu, Yansong Chua, Yuxiao Ning, Pengfu Liu, Yiwei Zhang, Tuoru Li, Guihua Wan, Zi-
> > > > jun Wan, Weidong Chen, and Shaomin Zhang. motorsrnn: A spiking recurrent neural network
> > > > inspired by brain topology for the effective and efficient decoding of cortical spike trains. Biomed-
> > > > ical Signal Processing and Control, 99:106745, January 2025. ISSN 1746-8094.
> > > >
> > > > [6] Rossbroich J, Gygax J, Zenke F. Fluctuation-driven initialization for spiking neural network training[J]. Neuromorphic Computing and Engineering, 2022, 2(4): 044016.

---

> > > > > ### Author Response · Authors · 2025-11-27
> > > > > **Official Comment by Authors**
> > > > >
> > > > > Dear Reviewer XSu9, We want to express our gratitude for dedicating your valuable time to review our paper meticulously and provide us with your valuable insights and feedbacks, which have significantly improved the quality of our work.
> > > > >
> > > > > As the Author-Review Discussion period is approaching the end, we would like to ensure that we have addressed all the concerns you raised. If you have any remaining questions or issues, we would be very happy to provide further clarification or make any necessary modifications.

---

### Author Response · Authors · 2025-11-20
**Summary of Author Response to All the Reviewers**

We extend our sincere appreciation to the reviewers and area chair for valuable and high-quality reviews. And we are encouraged that the reviewers found our work to have clear motivation  with a novel architecture, and they acknowledged its improved performance, low-power and lightweight design, and strong generalization potential supported by a thorough experimental design.

We revised the manuscript based on the constructive feedback and suggestions from the reviewers. Our key responses are summarized as below:


**overall architecture**

We have revised the description of the core structure of RAT SNN and renamed it **CST re-parametrized attention block** (from *re-parametrized channel attention block*) to avoid confusion between CST electrode “channels” and channel attention in the computer vision literature.

Furthermore, we have refined the structure of Section 3.2 by adding a detailed description of the CST re-parametrized attention block to highlight the architecture-level innovations tailored to our target scenario and to improve the readability of the paper.

Specifically, we emphasize: (i) the lightweight redesign of the prior spike-driven transformer , (ii) the parallel Conv+FC Q/K/V branches for capturing both local and global features under discontinuous electrode channel indices, and (iii) the introduction of a linear mapping on the spike-element-wise shortcut, which enables rescaling of the output layer of attention block.

**Ablation Study**

We have added Section 4.5, *Structural ablation study*, in the main text to compare our method with the spike transformer module and a classical recurrent SNN, and additional details are provided in Appendix E.1.1. The ablation results demonstrate the effectiveness of our architectural innovations

Following the suggestion of reviewer c3gQ, we have moved the ablation results on re-parametrization from the appendix into the main text to highlight the central role and effectiveness of this strategy in our method.

Due to space constraints, we placed the ablation study on R-LIF in Appendix E.1.2. The results show that combining R-LIF with the spike attention structure is effective.

**Stepwise training framework**

In Section 3.3, *Stepwise training framework*, we explicitly clarify the definition of cross-condition (encompassing multiple subjects and tasks) to avoid potential confusion for readers.


**Generalizing across conditions**
In Appendix C, *Evaluation details*, we have added Table 12 to provide the experimental results on the Perich dataset corresponding to Figure 5 and Figure 15, further highlighting the effectiveness of our method in generalizing to unseen conditions.


**Decoding accuracy comparison**

POYO remains a very strong ANN baseline for CST decoding and represents the top tier in terms of decoding accuracy. **Our goal is not to claim superiority over POYO in accuracy, but to achieve comparable performance**.
For example, on the NLB_RTT generalization task, our method outperforms POYO‑1 (the SOTA cross‑condition pretraining model), whereas on the more challenging NLB_MAZE task (Table 4), its decoding accuracy is lower than POYO‑1.
Since MAZE is the most complex task in our evaluation suite, this is consistent with the expectation that large‑scale ANN models are more likely to achieve higher decoding accuracy on sufficiently complex tasks.

What we would like to stress is that, under **comparable decoding accuracy**, our method provides **a substantial reduction in the computational cost (energy potential advantage) and an lightweight model size (on-chip memory potential advantage)**.
Moreover, RAT SNN produces outputs at every time step given the current spike input, while POYO predicts over fixed-length input segments; combined with the much shallower depth, RAT SNN also has **potential advantages for real-time inference** in iBMI applications, where efficiency and latency are as important as decoding accuracy.

**Discussion on the depth of SNN in CST decoding**

We have added Appendix E.4, *“Discussion on the depth of SNN in CST decoding task”*, to discuss the adverse effects of SNN depth on CST decoding, analyze the potential underlying causes, and provide additional preliminary experimental results as theoretical support for the architectural design of our model.

**Summary**

We thank all the reviewers again for the detailed and constructive review.
We are glad that all the reviewers have agreed with our contribution of light weight structure and low computational cost, and that most of the concerns are raised to insufficient description of the model architecture and the experimental evaluations about the decodingy accuracy comparison between ours and SOTA ANN methods (POYO).
We hope our additional description and the added experiments in this revision could address your concerns.
Let us know any remaining questions or concerns if you have.

---

### Comment · Area_Chair_S7En · 2025-11-25

Dear Reviewers,

This is a gentle reminder to please take a moment to review the authors’ rebuttal for the manuscript currently under your evaluation. Your timely feedback will help us proceed with the next steps in the review process.

Thank you for your time and assistance.

Best regards,
AC

---

### Author Response · Authors · 2025-11-30
**Summary of Author/Reviewer Discussion for AC**

**Dear AC,**

We understand the extra workload caused by the recent incident and appreciate your time. We summarize our rebuttal updates below to assist your assessment.

Before the Author/Reviewer Discussion, our paper received reviews from four reviewers [XSu9, a8mk, tbCB, c3gQ], with ratings of **[8, 6, 4, 6] (average 6)**.
We have responded carefully and thoroughly, **adding a substantial number of new experiments** and **refining the description of our architectural innovations**. We believe these updates have significantly strengthened the paper, particularly in the following aspects (**detailed in the section "Key Improvements" below**):
1. Architecture Details & Innovation [a8mk, tbCB, c3gQ],
2. Additional Ablation Study [c3gQ, tbCB],
3. Additional Generalization Analysis [a8mk, XSu9],
4. Decoding Performance Comparison to SOTA ANN (e.g., POYO) [XSu9, a8mk, tbCB].
5. Depth Analysis of SNN in CST decoding [tbCB].

### Reviewer Status
**1. Reviewer a8mk: Two rounds of Q&A, concerns addressed:**
We are glad that our additional experiments and clarifications effectively resolved the reviewer's concerns. In their final response, the reviewer acknowledged that the new results **"consolidate the claims of the paper"** and explicitly stated: “**I believe my concerns have been addressed** and I will change my score.” **(Score: 6 → 8)**

**2.  Reviewer c3gQ, concerns addressed:**
It is encouraging to see that the reviewer appreciated our detailed response. They confirmed that our response “...**effectively addressed my concerns**,” concluding with **"I have no further questions."** **(Score: 6)**

**3.  Reviewer XSu9, no response yet:**
We appreciate the reviewer's strong initial endorsement. Even without further discussion, they had already highlighted the value of our work, noting that: “Even if ANNs were to take a strong scoring lead in such a case compared to SNNs, this wouldn't reduce **the appeal of the presented method for real-time applications**.” **(Score: 8)**

**4.  Reviewer tbCB, no response yet:**
While the reviewer raised concerns that “the network architecture design lacks sufficient novelty,”  we are pleased that they also acknowledged that “the authors’ **innovations in training methodology and dataset integration are indeed appealing**.” **(Score: 4)**



Although reviewers XSu9 and tbCB did not provide further comments, **their key concerns substantially overlapped with those raised by reviewers a8mk and c3gQ, which have already been addressed by our new experiments and clarifications. We therefore expect that these updates would also satisfy XSu9 and tbCB**.

We understand that scores were reverted to prevent potential collusion. However, we believe the **score increase from Reviewer a8mk (6->8， 6->6.5 in Avg) in our case was the result of a constructive technical discussion and additional experiments**.

We are deeply grateful for the time and valuable feedback provided by you and the reviewers on this work, and we hope you will take these updates into account in your final recommendation.

**Sincerely,**

**All Authors**

---

> ### Author Response · Authors · 2025-11-30
> **Key Improvements**
>
> # Key Improvements
> To address the reviewers’ concerns, we made five major improvements during the rebuttal, which we summarize below.
> ### 1. Architecture Details & Innovation [a8mk, tbCB, c3gQ]
> We **revised Section 3.2 to detail the core structure of RAT SNN**, renaming it the CST re-parametrized attention block (formerly *re-parametrized channel attention block*) to avoid confusion between CST electrode “channels” and channel attention in the computer vision literature.
> This refinement highlights the **architecture‑level innovations tailored to our target scenario**, differentiates our design from classic spike transformer blocks [1–4], and significantly improves readability:
> 1. the lightweight redesign of the prior spike-driven transformer,
> 2. the parallel Conv+FC Q/K/V branches for capturing both local and global features under discontinuous electrode channel indices,
> 3. the introduction of a linear mapping on the spike-element-wise shortcut, which enables rescaling of the output layer of CST re-parametrized attention block.
>
> ### 2. Additional Ablation Study [c3gQ, tbCB]
> We have **extended the original ablation study** by adding comparisons with the classic spike transformer block and with regular recurrent SNNs using reparameterization (including both shallow and deep variants).
> We further conducted experiments on rescaling the output layer of the CST re‑parametrized attention block to validate the effectiveness of "introducing a linear mapping on the spike‑element‑wise shortcut", as well as ablations on the recurrent connection and synaptic dynamics in the R‑LIF layer described in the paper.
> We further conducted experiments on rescaling the output layer of the CST re‑parametrized attention block to validate the effectiveness of "introducing a linear mapping on the spike‑element‑wise shortcut". In addition, we performed ablations on the recurrent connection and synaptic dynamics in the R‑LIF layer described in the paper to further verify spiking-neural-level design choices.
>
> These results collectively demonstrate that the structural improvements in our method can effectively enhance decoding accuracy while preserving the lightweight nature of the model.
>
> ### 3. Additional Generalization Analysis [a8mk,XSu9]
> We added **additional sessions from Monkey C05_2025** to further evaluate the generalization performance of our method. On the currently available valid sessions, RAT SNN, trained with our stepwise training pipeline, can be rapidly and accurately adapted to unseen sessions from both seen and unseen conditions.
>
> Moreover, we **redesigned the tables** in the main text by marking generalization results with a superscript “+”. We also added comparative tables of test results at each stage in the appendix.
>
> ### 4. Decoding Performance Comparison to SOTA ANN (e.g., POYO) [XSu9, a8mk, tbCB]
> POYO remains a very strong ANN baseline for CST decoding and represents the top tier in terms of decoding accuracy. **Our goal is not to claim superiority over POYO in accuracy, but to achieve comparable performance**.
> For example, on the NLB_RTT generalization task, our method outperforms POYO, whereas on the more challenging NLB_MAZE task (Table 4), its decoding accuracy is lower than POYO.
> Since MAZE is the most complex task in our evaluation suite, this is consistent with the expectation that large‑scale ANN models are more likely to achieve higher decoding accuracy on sufficiently complex tasks.
>
> What we would like to stress is that, under **comparable decoding accuracy**, our method provides **a substantial reduction in the computational cost (energy potential advantage) and an lightweight model size (on-chip memory potential advantage)**.
> Moreover, RAT SNN produces outputs at every time step given the current spike input, while POYO predicts over fixed-length input segments; combined with the much shallower depth, RAT SNN also has **potential advantages for real-time inference** in iBMI applications, where efficiency and latency are as important as decoding accuracy.
>
> ### 5. Depth Analysis of SNN in CST decoding [tbCB]
> To address the concern regarding our shallow architecture, we **added Appendix E.4 with new experimental comparisons across different SNN depths**.
> Our results empirically demonstrate that increasing depth not only fails to improve decoding performance but in fact leads to a degradation for the CST task.
>
> Unlike natural images requiring deep hierarchies for spatial semantics, most neuromorphic benchmarks (including the CST task) typically involve “event streams + relatively simple spatial structure,” where the key to decoding lies primarily in temporal patterns.
> **Our shallow design effectively captures these rhythms while avoiding the optimization difficulties of deep SNNs** [5], thus achieving high decoding accuracy with significantly lower latency and energy cost.

---

> ### Author Response · Authors · 2025-11-30
> **Key Improvements Ref**
>
> [1] Zhou Z, Zhu Y, He C, et al. Spikformer: When spiking neural network meets transformer[J]. arXiv preprint arXiv:2209.15425, 2022.
>
> [2] Zhang J, Tang L, Yu Z, et al. Spike transformer: Monocular depth estimation for spiking camera[C]//European Conference on Computer Vision. Cham: Springer Nature Switzerland, 2022: 34-52.
>
> [3] Yao M, Hu J K, Hu T, et al. Spike-driven transformer v2: Meta spiking neural network architecture inspiring the design of next-generation neuromorphic chips[J]. arXiv preprint arXiv:2404.03663, 2024.
>
> [4] Yao M, Hu J, Zhou Z, et al. Spike-driven transformer[J]. Advances in neural information processing systems, 2023, 36: 64043-64058.
>
> [5] Rossbroich J, Gygax J, Zenke F. Fluctuation-driven initialization for spiking neural network training[J]. Neuromorphic Computing and Engineering, 2022, 2(4): 044016.

---

### Meta-Review · Area_Chair_g3KB · 2026-01-09

**Summary:**

This paper proposes RAT-SNN, a re-parameterized self-attention spiking neural network (SNN) designed for decoding in intracortical BCI settings, with an explicit emphasis on lightweight deployment, and generalization across sessions, subjects, and tasks. Across the review process, the work received mostly positive evaluations, with reviewers highlighting the clear motivation, careful experimental design, and the importance of efficient SNN-based alternatives to large ANN decoders. The core contribution lies in a task-specific, shallow spike-attention architecture coupled with a stepwise cross-condition pretraining framework, which achieves decoding accuracy comparable to strong ANN baselines (e.g., POYO) while substantially reducing parameter count and computational cost.

The rebuttal and discussion period significantly strengthened the paper. The authors added extensive new experiments, including additional ablations to compare the classic spike transformer block and with regular recurrent SNNs using reparameterization, rescaling the output layer of the CST re‑parametrized attention block, as well as ablations on the recurrent connection and synaptic dynamics in the R‑LIF layer in their model. They also added additional sessions from Monkey C05_2025 to test further generalization, and added comparisons with POYO, where they found comparable performance with a much lighterweight design. Multiple reviewers explicitly confirmed that their concerns were fully addressed, with at least one reviewer upgrading their score.

Overall, this work represents a solid and timely contribution to brain-inspired neural decoding. The thorough rebuttal, added experiments, and constructive engagement with reviewers substantially improved clarity and deepened the rigor of the evaluations. Given all this, I would recommend acceptance.

A note to the authors: Please take care to add in the new results, suggested changes, and also add a discussion to recent related work on SNN for BMI on these same datasets by Mentzelopoulos et al. presented at NeurIPS 2025.

**Reviewer Concerns:**

The authors did a great job of addressing the reviewers' concerns.

**Reviewer Scores:**

XSu9 (8)
A8mk (6 -> 8)
tbCB (4 -> 6)
c3gQ (6)

---

### Decision · Program_Chairs · 2026-01-26

Accept (Poster)